# Age and diet shape the genetic architecture of body weight in diversity outbred mice

Kevin M Wright[1†], Andrew G Deighan[2], Andrea Di Francesco[1], Adam Freund[1], Vladimir Jojic[1], Gary A Churchill[2]*, Anil Raj[1]*†

[1]Calico Life Sciences LLC, South San Francisco, United States; [2]The Jackson Laboratory, Bar Harbor, United States

**Abstract** Understanding how genetic variation shapes a complex trait relies on accurately quantifying both the additive genetic and genotype–environment interaction effects in an age-dependent manner. We used a linear mixed model to quantify diet-dependent genetic contributions to body weight measured through adulthood in diversity outbred female mice under five diets. We observed that heritability of body weight declined with age under all diets, except the 40% calorie restriction diet. We identified 14 loci with age-dependent associations and 19 loci with age- and diet-dependent associations, with many diet-dependent loci previously linked to neurological function and behavior in mice or humans. We found their allelic effects to be dynamic with respect to genomic background, age, and diet, identifying several loci where distinct alleles affect body weight at different ages. These results enable us to more fully understand and predict the effectiveness of dietary intervention on overall health throughout age in distinct genetic backgrounds.

## Editor's evaluation

This is an outstanding dissection of the genetic architecture of body weight at the genome-wide level across time and across environments. The use of a multiparental mouse population permits high-resolution mapping. The statistical analyses are advanced, leveraging new models, as well as tools developed specifically for this mouse population. The corresponding results are presented in nice and informative figures.

*For correspondence:
gary.churchill@jax.org (GAC);
anil@calicolabs.com (AR)

†These authors contributed equally to this work

## Introduction

Quantifying the contributions of genetic and environmental factors to population variation in an age-dependent phenotype is critical to understanding how phenotypes change over time and in response to external perturbations. The identification of genetic loci that are associated with a complex trait in an age- and environment-dependent manner allows us to elucidate the dynamics and context dependence of the genetic architecture of the trait and facilitates trait prediction. For health-related traits, these genetic loci may also facilitate greater understanding and prediction of age-related disease etiology, which is an important step to genetically or pharmacologically manipulating these traits to improve health.

Standard approaches used to identify genetic loci associated with quantitative traits can be confounded by nonadditive genetic effects such as genotype–environment (G×E) and genotype–age (G×A) interactions (*Robinson et al., 2017*), contributing to the 'missing heritability' of quantitative traits (*Sul et al., 2016*). The linear statistical models routinely used in genetic mapping analyses do not account for variation in population structure between environments and polygenicity in G×E

**eLife digest** Body weight is one trait influenced by genes, age and environmental factors. Both internal and external environmental pressures are known to affect genetic variation over time. However, it is largely unknown how all factors – including age – interact to shape metabolism and bodyweight.

Wright et al. set out to quantify the interactions between genes and diet in ageing mice and found that the effect of genetics on mouse body weight changes with age. In the experiments, Wright et al. weighed 960 female mice with diverse genetic backgrounds, starting at two months of age into adulthood. The animals were randomized to different diets at six months of age. Some mice had unlimited food access, others received 20% or 40% less calories than a typical mouse diet, and some fasted one or two days per week.

Variations in their genetic background explained about 80% of differences in mice's weight, but the influence of genetics relative to non-genetic factors decreased as they aged. Mice on the 40% calorie restriction diet were an exception to this rule and genetics accounted for 80% of their weight throughout adulthood, likely due to reduced influence from diet and reduced interactions between diet and genes. Several genes involved in metabolism, neurological function, or behavior, were associated with mouse weight.

The experiments highlight the importance of considering interactions between genetics, environment, and age in determining complex traits like body weight. The results and the approaches used by Wright et al. may help other scientists learn more about how the genetic predisposition to disease changes with environmental stimuli and age.

interactions. Population structure can substantially increase the false-positive rate when testing for G×E associations (*Moore et al., 2019*). Furthermore, not accounting for polygenic G×E interactions has the potential to incorrectly estimate the heritability of quantitative traits in the context of specific environments (*Sul et al., 2016*). To address these limitations, recent efforts have generalized standard linear mixed models (LMMs) with multiple variance components that allow for polygenic G×E interactions and environment-dependent residual variation (*Sul et al., 2016*; *Runcie and Crawford, 2019*; *Moore et al., 2019*; *Dahl et al., 2020*). Moreover, these LMMs substantially increase the power to discover genomic loci that are associated with phenotype in both an environment-independent and environment-dependent manner.

In this study, we used an LMM to investigate the classic quantitative trait, body weight, in a large population of diversity outbred (DO) mice. Body weight was measured longitudinally from early development to late adulthood, before and after the imposition of dietary intervention at 6 months of age. We expect diet and age to be important factors affecting body weight and growth rate; however, it remains to be determined how these factors will interact with genetic variation to shape growth. Two early studies found significant genetic correlations for body weight and growth rate during the first 10 weeks of mouse development, which supported the hypothesis that growth rates during early and late development were affected by pleiotropic loci (*Cheverud et al., 1996*; *Riska et al., 1984*). Subsequent experiments found that the heritability of body weight increased monotonically with age throughout development: from 29.3% to 76.1% between 1 and 10 weeks of age (*Cheverud et al., 1996*), from 6% to 24% between 1 and 16 weeks of age (*Gray et al., 2015*), and from 9% to 32% between 5 and 13 weeks of age (*Parker et al., 2016*). The heritability of growth rate also varied with age, but exhibited a peak of 24% at 3 weeks of age and then declined to nearly 4% at 16 weeks of age (*Gray et al., 2015*). The strength of association and effect size of QTLs for body weight and growth rate were specific to early or late ages and were inconsistent with the hypothesis that pleiotropic alleles affect animal size at early and late developmental stages (*Cheverud et al., 1996*; *Gray et al., 2015*). While these results are well supported, their interpretation is somewhat limited because body weight measurements ceased at young ages and significant QTLs encompassed fairly large chromosomal regions. Given these limitations, we were motivated to ask two questions: Will fine-mapping to greater resolution reveal single genes that function at either early or late developmental stages, or reveal multiple genes in tight linkage with variable age-specific effects? How will the effect of these loci change at later ages and under different diets?

We expect the interaction of dietary interventions, such as calorie restriction (CR) or intermittent fasting (IF), with genetics to greatly impact the body weight trajectories of mice. Researchers have observed genotype-dependent reductions in body weight in the 7–50 weeks after imposing a 40% CR diet, and variation in heritability of this trait with age from 42% to 54% (*Rikke et al., 2006*). A second study subjected a large genetic-mapping population to dietary intervention and identified multiple loci with significant genetic and genotype–diet interaction effects on body weight at 2–6 months of age (*Vorobyev et al., 2019*). These studies identified substantial diet- and age-dependent genetic variance for body weight in mice, similar to what has been found in humans (*Robinson et al., 2017*; *Couto Alves et al., 2019*; *Wang et al., 2019*). It, however, remains to be determined how the

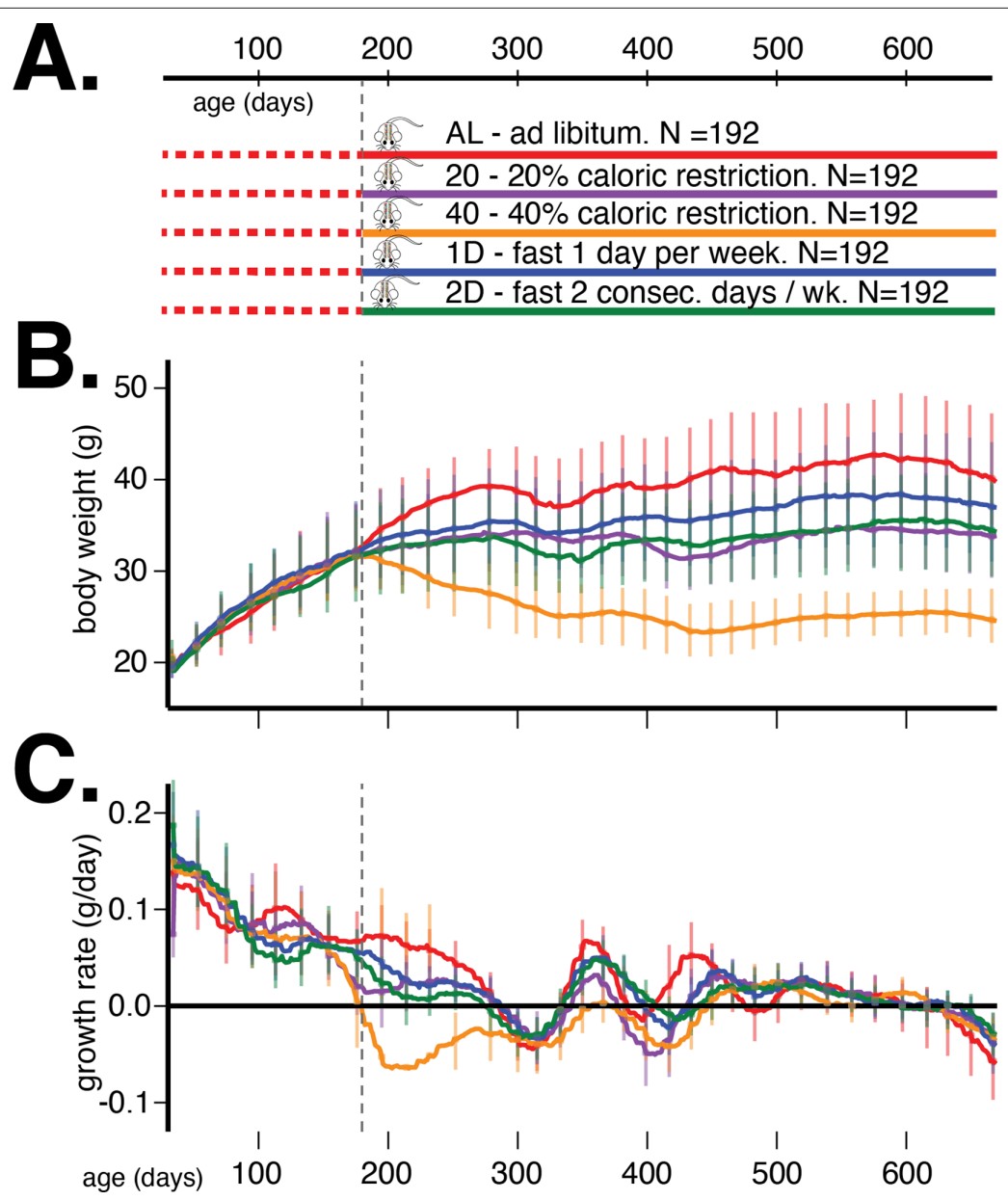

**Figure 1.** Study design and body weight data summaries. (**A**) Outline of study design. (**B**) Median (interquartile range) body weight in grams and (**C**) median (interquartile range) growth rate in grams per day for five dietary treatments from 60 to 660 days of age. Vertical gray dotted line denotes the onset of dietary intervention at 180 days of age.

The online version of this article includes the following figure supplement(s) for figure 1:

**Figure supplement 1.** Raw phenotype measurements.

contribution of specific genetic loci to body weight changes in response to different dietary interventions and whether the effects observed in younger mice are indicative of the maintenance of body weight in adult mice.

In order to address these questions, we measured the body weight of multiple cohorts of genetically diverse mice from the DO population (*Svenson et al., 2012*) from 60 to 660 days of age (*Figure 1A*). At 180 days of age, we randomized mice by body weight and assigned each mouse to one of five dietary regimes – ad libitum (AL), 20% and 40% daily CR, and 1 or 2 days per week IF. Longitudinal measurements of body weight in these DO mice allowed us to discover the age-dependent genetic determinants of body weight and growth rate in the context of different dietary interventions.

In the following sections, we first describe the study design and collection of the genetic and body weight measurements. Second, we motivate the use of the gene–environment mixed model (G×EMM) (*Dahl et al., 2020*) to model genetic associations in these data and give an overview of our analyses. We next use this model to identify genetic loci having additive or genotype–diet interaction effects on body weight. We fine-map candidate loci and determine the scope of pleiotropy for age- and diet-specific effects. We find many, but not all, loci are associated with body weight in a narrow age range and localize to small genomic regions, in some cases to single genes. We utilize the full genome sequence of the DO founders and external chromatin accessibility data to further narrow the genomic regions to a small number of candidate variants at each locus. Interestingly, many diet-specific body weight loci localize genes implicated in neurological function and behavior in mice or humans.

## Results and discussion
### Study design and measurements

The DO house mouse (*Mus musculus*) population was derived from eight inbred founder strains and is maintained at Jackson Labs as an outbred heterozygous population (*Svenson et al., 2012*). This study contains 960 female DO mice, sampled at generations: 22–24 and 26–28. There were two cohorts per generation for a total of 12 cohorts and 80 animals per cohort. Enrollment occurred in successive quarterly waves starting in March 2016 and continuing through November 2017.

A single female mouse per litter was enrolled into the study after wean age (3 weeks old), so that no mice in the study were siblings and maximum genetic diversity was achieved. Mice were housed in pressurized, individually ventilated cages at a density of eight animals per cage (cage assignments were random). Mice were subject to a 12 hr:12 hr light:dark cycle beginning at 06:00 hr. Animals exit the study upon death. All animal procedures were approved by the Animal Care and Use Committee at The Jackson Laboratory.

From enrollment until 6 months of age, all mice were on an AL diet of standard rodent chow 5KOG from LabDiet. At 6 months of age, each cage of eight animals was randomly assigned to one of five dietary treatments, with each cohort equally split between the five groups (N = 192/group): AL, 20% CR (20), 40% CR (40), 1 day per week fast (1D), and 2 days per week fast (2D) (see *Figure 1A*). In a previous internal study at the Jackson Laboratory, the average food consumption of female DO mice was estimated to be 3.43 g/day. Based on this observation, mice on 20% CR diet were given 2.75 g/mouse/day and those on 40% CR diet were given 2.06 g/mouse/day. Food was weighed out for an entire cage of eight. Observation of the animals indicated that the distribution of food consumed was roughly equal among all mice in a cage across diet groups.

Mice on AL diet had unlimited food access; they were fed when the cage was changed once a week. In rare instances when the AL mice consumed all food before the end of the week, the grain was topped off midweek. Mice on 20% and 40% CR diets were fed daily. We gave them a triple feeding on Friday afternoon to last till Monday afternoon. As the number of these mice in each cage decreased over time, the amount of food given to each cage was adjusted to reflect the number of mice in that cage. Fasting was imposed weekly from Wednesday noon to Thursday noon for mice on 1D diet and Wednesday noon to Friday noon for mice on 2D diet. Mice on 1D and 2D diets have unlimited food access (similar to AL mice) on their nonfasting days.

From enrollment until 660 days of age, each animal underwent a number of phenotypic assays to assess a wide range of physiological health parameters. These included two 7-day stints in Sable Systems Promethion metabolic cages at 120 and 390 days of age, two blood draws at 160 and 480 days of age, and one of each of the following challenge-based assays at 300–330 days of age:

3-day stint in a cage with wheel, rotarod, grip strength, dual-energy X-ray absorptiometry, echocardiogram, Y-maze spontaneous alternation, and acoustic startle.

## Body weight measurements

Body weight was measured once every week for each mouse throughout its life. The body weight measurements for this analysis were collated on February 1, 2020, at which point 941 mice (98%) had measurements at 180 days, 890 (93%) at 365 days, 813 (85%) at 550 days, and 719 (75%) at 660 days. For these analyses, we included all body weight measures for each mouse up to 660 days of age. We smoothed out measurement noise, either due to errors in measurement or swaps in assigning measurements to mice, using an $\ell_1$ trend filtering algorithm (*Kim et al., 2009*), which calculates a piecewise linear trend line for body weight for each mouse over its measurement span. The degree of smoothing was learned by minimizing the error between the predicted fit and measurements at randomly held-out ages across all mice. In the rest of this article, *body weight* and *growth rate* refer to the predicted fits from $\ell_1$ trend filtering.

We present the average trends in body weight and growth rate, stratified by dietary intervention, in *Figure 1B and C*, respectively. The body weight and growth rate trends without $\ell_1$ trend filtering are presented in *Figure 1—figure supplement 1*. The most prominent observation from these trends is that dietary intervention contributes the most to variation in body weight in this mouse population. After accounting for this source of variation, there remain substantial and different quantities of variation in body weight trends within the different dietary interventions, suggesting the existence of heteroscedastic noise or plausible G×D interaction effects on body weight trajectories.

## Genotype measurements

We collected tail clippings and extracted DNA from 954 animals (http://agingmice.jax.org/protocols). Samples were genotyped using the 143,259-probe GigaMUGA array from the Illumina Infinium II platform (*Morgan et al., 2015*) by NeoGen Corp. (genomics.neogen.com/). We evaluated genotype quality using the R package: qtl2 (*Broman et al., 2019*). We processed all raw genotype data with a corrected physical map of the GigaMUGA array probes (https://kbroman.org/MUGAarrays/muga_annotations.html). After filtering genetic markers for uniquely mapped probes, genotype quality, and a 20% genotype missingness threshold, our dataset contained 110,807 markers.

We next examined the genotype quality of individual animals. We found seven pairs of animals with identical genotypes, which suggested that one of each pair was mislabeled. We identified and removed a single mislabeled animal per pair by referencing the genetic data against coat color. Next, we removed a single sample with missingness in excess of 20%. The final quality assurance analysis found that all samples exhibited high consistency between tightly linked markers: log odds ratio error scores were less than 2.0 for all samples (*Lincoln and Lander, 1992*). The final set of genetic data consisted of 946 mice.

For each mouse, starting with its genotypes at the 110,807 markers and the genotypes of the 8 founder strains at the same markers, we inferred the founders-of-origin for each of the alleles at each marker using the R package: qtl2 (*Broman et al., 2019*). This allowed us to test directly for association between founder-of-origin and phenotype (rather than allele dosage and phenotype, as is commonly done in QTL mapping) at all genotyped markers. Using the founder-of-origin of consecutive typed markers and the genotypes of untyped variants in the founder strains, we then imputed the genotypes of all untyped variants (34.5 million) in all 946 mice. Targeted association testing at imputed variants allowed us to fine-map QTLs to a resolution of 1–10 genes.

## Motivating models for environment-dependent genetic architecture

Genome-wide QTL analyses in model organisms over the last decade have predominantly employed linear mixed models (e.g., EMMA *Kang et al., 2008*), FastLMM [*Lippert et al., 2011*], GREML [*Yang et al., 2011*], GEMMA [*Zhou and Stephens, 2012*], and LDSC [*Bulik-Sullivan et al., 2015*], expanding on the heuristic that samples sharing more of their genome have more correlated phenotypes than genetically independent samples. We found that the distributions of covariances in body weight, measured at 500 days of age, between animal pairs within the AL treatment were nearly indistinguishable when we partition pairs into high-kinship (>0.2) and low-kinship groups (*Figure 2*, no significant separation between solid and dashed red lines). However, animal pairs in the 40% CR treatment

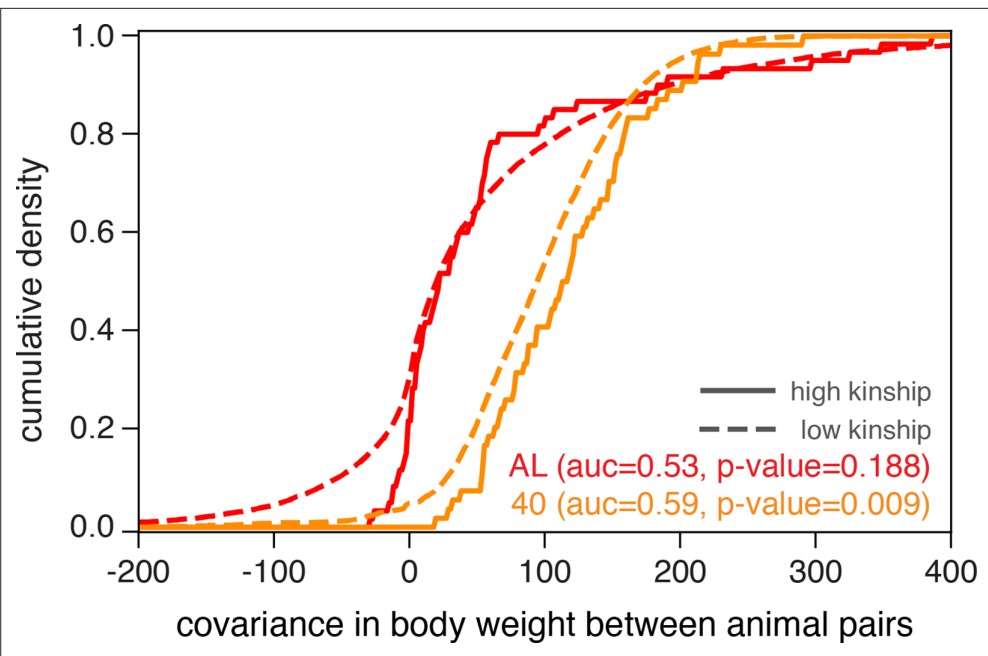

**Figure 2.** Illustrating diet-dependent association between genetic and phenotypic similarities. Phenotypic divergence between animal pairs is quantified by the covariance in body weight at 500 days of age. We plot the cumulative density of body weight covariance for all pairs of animals in the ad libitum (AL) and 40% calorie restriction (CR) dietary treatments, partitioned into high-kinship (>0.2) or low-kinship groups. We test for separability between the high- and low-kinship distributions using the Mann–Whitney $U$ test, and report p-values from this test. The reported area under the curve (AUC) is a standard transformation of the Mann-Whitney U test statistic.

exhibited significantly lower covariance in body weight in the low-kinship group compared to the high-kinship group (*Figure 2*, significant separation between solid and dashed orange lines). This observation suggests that the genetic contribution to body weight is different in distinct dietary environments. This observation also motivates the use of recently developed generalized linear mixed models to conduct genome-wide QTL analysis because they more fully account for environment-dependent genetic variances, reduce false-positive rates, and increase statistical power (*Yang et al., 2011*; *Sul et al., 2016*; *Runcie and Crawford, 2019*; *Moore et al., 2019*; *Dahl et al., 2020*).

## Simulations

To evaluate the accuracy of G×EMM and compare it against standard linear mixed models like EMMA (*Kang et al., 2008*), we simulated phenotypic variation under a broad range of values for proportion of phenotypic variance explained by genetics (PVE) in each of two environments. For all simulations, we fixed the total sample size to $N = 946$ and used the observed kinship matrix for the 946 DO mice in this study. For these simulations, the environment-specific genetic contribution $\mathrm{PVE}_e$, environment-specific noise $\sigma_e^2$, and the relative distribution of samples between environments are the tunable parameters. In order to explore how the two models behaved under a wide range of conditions, we varied $\mathrm{PVE}_e \in \{0.05, 0.10, \ldots, 0.95\}$, fixed $\sigma_{e1}^2 = \sigma_{e2}^2 = 1.0$, and assigned samples to environments either at a 1:1 ratio or a 4:1 ratio (i.e., a total of 114 parameter values). For each setting of the tunable parameter values, we ran 50 replicate simulations and in each simulation, we randomly assigned samples to one of the two environments. Given the environment assignments, kinship matrix, and the set of tunable parameter values for each simulation, we computed the environment-specific genetic variance using *Equation 13*, *Equation 14*, and *Equation 15*. Next, we generated the vector of phenotypes from a multivariate normal distribution with zero mean and covariance structure dependent upon the observed kinship matrix, environment assignments, tunable parameter settings, and the environment-specific genetic variance for each environment. Using the simulated phenotypes, we

computed estimates of $\text{PVE}_{tot}$ under both the EMMA and G×EMM models and estimates of $\text{PVE}_e$ under the G×EMM model.

We first evaluated the accuracy of $\text{PVE}_{tot}$ estimated from the two models and found that the G×EMM model estimated total PVE with little bias and lower variance compared to EMMA in nearly all simulations across the suite of parameter combinations that we investigated (*Figure 3A and B*). In particular, we found that EMMA substantially underestimated the total PVE in comparison to G×EMM when samples were equally distributed between environments and there was substantial difference in PVE or noise between environments. This is because $\text{PVE}_e$ will have a greater impact on $\text{PVE}_{tot}$ when both environments are more equally represented in the study sample.

Next, we examined the sensitivity and specificity of the G×EMM model to quantify environment-specific genetic contribution to phenotypic variance. We found that when samples were equally distributed between environments, G×EMM accurately estimates $\text{PVE}_e$ under a wide range of parameter values (*Figure 3C*). When there was a skew in the number of samples between environments, the $\text{PVE}_e$ estimate for the environment with the larger sample size was as accurate as when there was no skew in sample size (similar to *Figure 3C*, result not shown). In contrast, the $\text{PVE}_e$ estimate for the environment with the smaller sample size was more variable and the median $\text{PVE}_e$ estimates were consistently lower than the true $\text{PVE}_e$ (*Figure 3D*). Finally, we observed that increasing the environment-specific noise terms, $\sigma_{e1}^2$ and $\sigma_{e2}^2$, increased the variance of our estimates of $\text{PVE}_e$ but introduced little bias (results not shown).

Overall, the G×EMM model outperforms the EMMA model in estimating total PVE and shows little bias in estimating environment-specific PVE across a broad range of scenarios relevant to our study.

## Overview of analyses

Starting with body weight measurements in 959 mice from 30 to 660 days of age, and founder-of-origin alleles inferred at 110,807 markers in 946 mice, we first quantified how the heritability of body weight and growth rate changes with age and between dietary contexts. We used the G×EMM model (described in detail in 'Materials and methods') to account for both additive environment-dependent fixed effects and polygenic gene–environment interactions. We considered two different types of environments: diet and generation. The five diet groups were applied from 180 to 660 days and the 12 generations were applied from 30 to 660 days. Next, we performed genome-wide QTL mapping for body weight at each age independently, testing for association between body weight and the inferred founder-of-origin at each genotyped marker. For ages 180–660 days, we additionally tested for association between body weight and the interaction of diet and founder-of-origin at each marker. We computed p-values using a sequential permutation procedure (*Besag and Clifford, 1991*; *Shim and Stephens, 2015*) at each variant for each of the additive and interaction tests and used these to assign significance (*Abney, 2015*). Finally, for each significant locus, we performed fine-mapping to identify the putative causal variants and founder alleles driving body weight and underlying functional elements (genes and regulatory elements) to ascertain the possible mechanisms by which these variants act.

## PVE of body weight across age and diet

First, we quantified the overall contribution of genetics to variation in body weight and, importantly, how this contribution changed with age. We applied both EMMA and G×EMM to body weight estimated every 10 days. Since the mice from each generation cohort were measured at the same time every week, we used generation as a proxy for the shared environment that mice are exposed to as part of the study design. We accounted for generation-specific fixed effects (α in *Equation 1*) in both models and genotype-generation random effects (γ) in G×EMM. For ages after dietary intervention ($\geq$ 180 days), we additionally accounted for diet-specific fixed effects in both models and genotype-diet random effects in G×EMM. We estimated the variance components in the model at each age independently and computed the total and diet-dependent PVE using *Equation 7* and *Equation 13*.

In *Figure 4A*, we observed that the PVE of body weight steadily increased during early adulthood and up to 180 days of age, when dietary intervention was imposed. The G×EMM model estimates a higher PVE than the EMMA model during this age interval because the former model specifically accounts for polygenic genotype-generation effects. Following dietary intervention, PVE decreased in four of the five dietary groups; the one exception was the 40% CR group, which maintained a high PVE

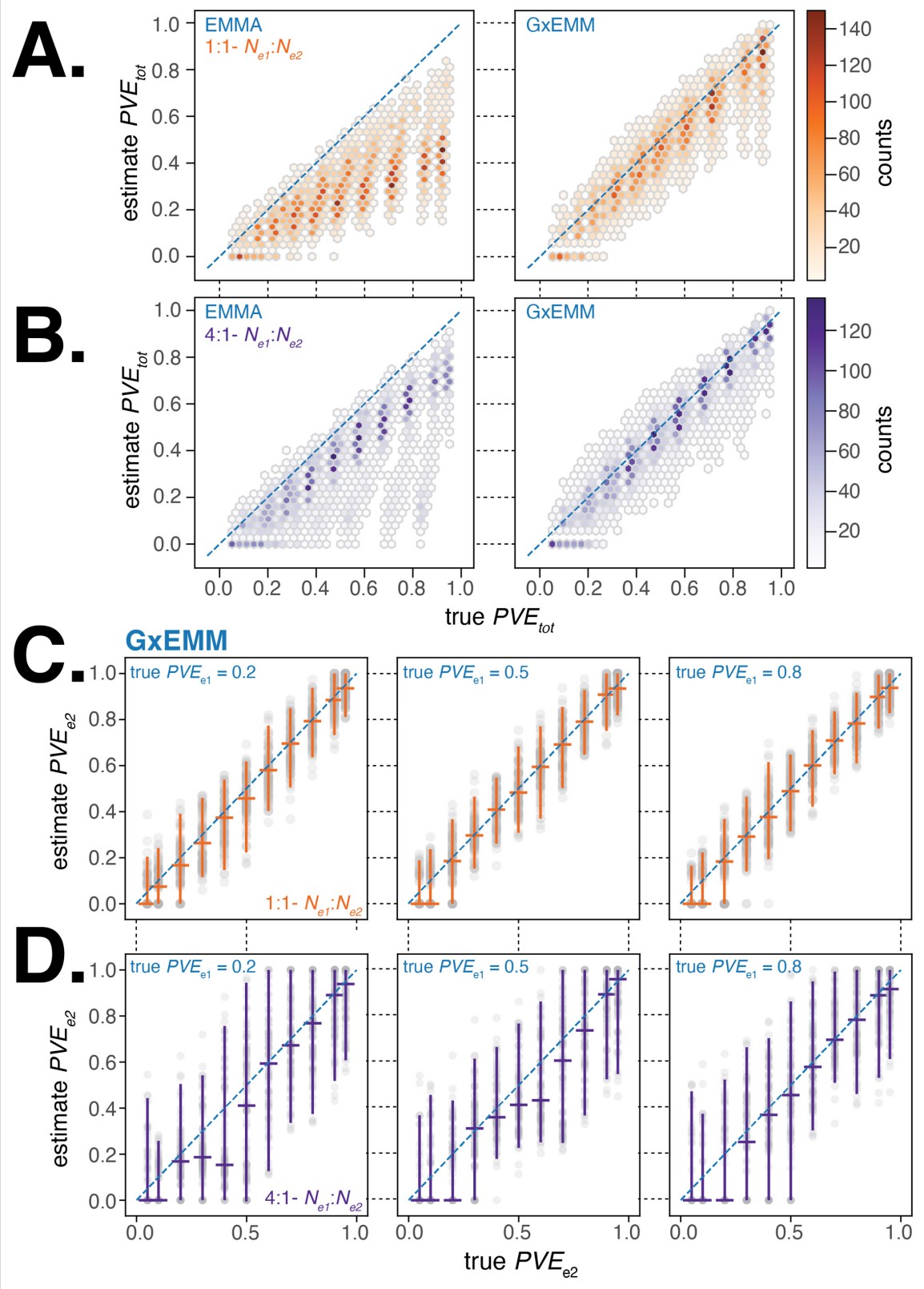

**Figure 3.** Evaluating gene–environment mixed mode (G×EMM) and EMMA using simulated datasets. (**A**). Comparison of true $\mathbf{PVE}_{tot}$ to that estimated from EMMA (left panel) and G×EMM (right panel). Simulations were run with an equal number of samples in each environment ($N_{e1} = N_{e2}$) and with the same value for the environment-specific error terms ($\sigma_{e1}^2 = \sigma_{e2}^2 = 1$). (**B**). Same as (**A**) but with a 4:1 ratio of samples between environments ($N_{e1} = 4 \times N_{e2}$). (**C**). Plot of the true $\mathbf{PVE}_{e2}$ vs. the G×EMM estimate of $\mathbf{PVE}_{e2}$. Gray points are the results of individual simulations, orange lines

*Figure 3 continued on next page*

*Figure 3 continued*

denote the median and 95% interquartile range (IQR). The three panels differ in the true proportion of phenotypic variance explained by genetics (PVE) specific to environment 1, $\mathrm{PVE}_{e1} \in \{0.2, 0.5, 0.8\}$. Simulations were run with an equal number of samples in each environment ($N_{e1} = N_{e2}$) and with the same value for the environment-specific error terms ($\sigma_{e1}^2 = \sigma_{e2}^2 = 1$). (**D**). Same as (**C**) but with a 4:1 ratio of samples between environments ($N_{e1} = 4 \times N_{e2}$).

($\mathrm{PVE}_{40} \approx 0.8$) (***Figure 4A***) and low total phenotypic variance (***Figure 4—figure supplement 3A***, top-right panel) from 180 to 660 days of age. In contrast, the AL group had the lowest PVE and greatest total phenotypic variance in the same age range. Notably, $\ell_1$ trend filtering of the raw measurements proved useful in quantifying smoothly varying trends in the PVE of body weight and growth rate, and improving our estimates of the PVE of growth rate by reducing the effect of measurement noise (***Figure 4—figure supplement 1***). Moreover, these results were robust to variation in the genetic data used to calculate the kinship matrix and to survival bias at 660 days of age.

The kinship matrix used for estimating these PVE values was computed based on the founder-of-origin of marker variants (***Aylor et al., 2011***). When using kinship estimated using biallelic marker genotypes (as is commonly done in genome-wide association studies [GWAS]), we observed largely similar trends in PVE; however, differences in PVE between diets after 400 days were harder to discern due to much larger standard errors for the estimates (***Figure 4—figure supplement 3A***, left panels). To test for bias or calibration errors in our PVE estimates, we randomly permuted the body weight trends between mice in the same diet group and recalculated the total and diet-dependent PVE values. Consistent with our expectations, PVE dropped to nearly zero for the permuted dataset (***Figure 4—figure supplement 3B***, left panel), indicating that the PVE estimates are well-calibrated. Finally, to evaluate the contribution of survival bias to our estimates, we recomputed PVE at all ages after restricting the dataset to mice that were alive at 660 days. We observed PVE estimates largely similar to those computed from the full dataset, suggesting very little contribution of survival bias to our estimates (***Figure 4—figure supplement 3C***).

When using kinship computed from genotypes, we note that the PVE in the permuted dataset does not drop to zero, suggesting that genotype-based kinship includes some background relatedness that can explain some of the phenotypic variation even after permuting the labels (***Figure 4—figure supplement 3B***, right panel). The genotype-based kinship includes two components: the genetic sharing between the founder strains and the genetic sharing arising from the breeding strategy used to develop the DO panel. Randomly permuting the phenotype labels breaks the link between phenotypic similarity and the latter component of genotype-based kinship, but does not effectively break the link with the former component. Kinship computed using the founder allele probabilities explicitly includes only the latter component; this component explains little phenotypic variation once the phenotype labels are randomly permuted.

Next, we quantified the age-dependent contribution of genetics to variation in growth rate, enabled by the dense temporal measurement of body weight. As before, we applied EMMA and G×EMM to growth rate estimated every 10 days. In ***Figure 4B***, we observed that PVE of growth rate increases rapidly during early adulthood, and then decreases to negligible values around 240 days of age. In contrast to body weight, PVE of growth rate is substantially lower at all ages, and there is little divergence in PVE across diet groups for most ages. Notably, the decrease and subsequent increase in growth rate PVE coincide with specific metabolic, hematological, and physiological phenotyping procedures that these mice underwent at specific ages as part of the study (***Figure 1—figure supplement 1***). Due to lower values and greater variance in PVE of growth rate with age, we focus on body weight throughout the rest of the article.

In summary, the 40% CR intervention produced the greatest reduction in average body weight and maintained a high PVE after dietary intervention. This is because the genetic variance in body weight remained relatively high and the environmental variance remained relatively low throughout this interval. In contrast, body weight PVE steadily decreased with age in each of the four less restrictive diets, which was due to a steady increase in noise (nongenetic variance) and not a decrease in the genetic variance of body weight (***Figure 4—figure supplement 3A***). Even though the genetic variance in body weight is nearly constant across diets from 180 to 660 days of age, the effect of specific variants may change with age.

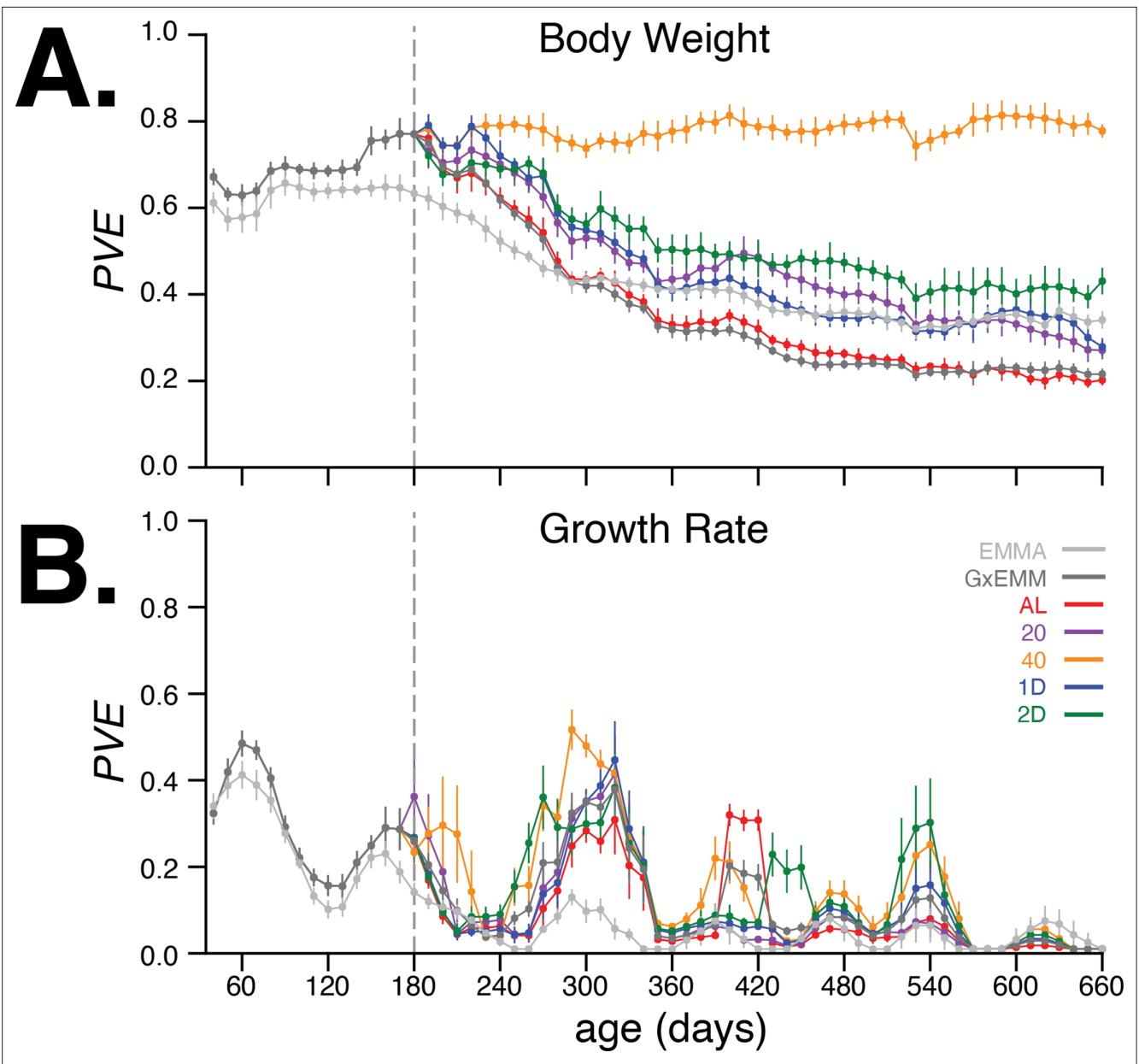

**Figure 4.** Proportion of phenotypic variance explained by genetics. (**A**) Body weight proportion of phenotypic variance explained by genetics (PVE) (± SE) for 30–660 days of age. Total PVE estimates are derived from the EMMA (light gray) and gene–environment mixed mode (G×EMM) (dark gray) models. Diet-dependent PVE values were derived from the G×EMM model. Dotted vertical line at 180 days depicts the time at which all animals were switched to their assigned diets. (**B**) Growth rate PVE; details the same as (**A**).

The online version of this article includes the following figure supplement(s) for figure 4:

**Figure supplement 1.** Proportion of phenotypic variance explained by genetics (PVE) using raw measurements.

**Figure supplement 2.** Proportion of phenotypic variance explained by genetics (PVE) overlayed with phenotyping.

**Figure supplement 3.** Sensitivity of proportion of phenotypic variance explained by genetics (PVE).

**Figure supplement 4.** Proportion of phenotypic variance explained by genetics (PVE) with no constraints on variance components.

**Figure supplement 5.** Proportion of phenotypic variance explained by genetics (PVE) comparing full and diagonal $\Omega$.

**Figure supplement 6.** Proportion of phenotypic variance explained by genetics (PVE) decomposition.

## Genome-wide QTL analysis of body weight across age and diet

We sought to identify loci significantly associated with body weight in a diet-dependent and age-dependent manner. To this end, we tested for association between the inferred founder-of-origin of each typed variant and body weight at each age independently. We note that body weight measurements taken at different ages are not independent; that is, we may detect a locus at a specific age if it has small effects on body weight acting over a long period of time or a large effect on body weight, resulting in rapid bursts in growth. Thus, a locus identified as significant at a specific age indicates its cumulative contribution to body weight at that age.

We identified 29 loci significantly associated with body weight at any age in the additive or interaction models using a genome-wide Bonferroni-corrected $p \leq 10^{-4}$ threshold for both the additive and the genotype–diet interaction tests (*Figure 5—figure supplement 1*). Additionally, we report suggestive genotype–diet interaction QTLs using a weaker threshold of $p \leq 10^{-3}$. The Bonferroni correction was computed based on the expected number of linkage-disequilibrium (LD) blocks in our dataset; as a proxy for the number of LD blocks, we used the number of top eigenvalues of the LD matrix between genotyped variants that explained at least 90% of the variance across markers. This threshold also corresponds to a 10% variant-level expected false discovery rate, using the Benjamini–Hochberg method (*Benjamini and Hochberg, 1995*). In order to focus on highly significant associations, we also used the number of eigenvalues explaining 99.5% of the variance across markers (*Gao et al., 2010*), resulting in a more stringent threshold of $p \leq 1.1 \times 10^{-5}$ (see *Table 1*). Notably, these thresholds hold for the interaction tests as well because we have only one test per marker. Using all biallelic variants imputed from the complete genome sequences of the eight DO founder strains (*Keane et al., 2011*), we retested the genetic association for the 29 candidate loci with the additive and interaction models. Specifically, we tested for association between all imputed genotype variants and body weight, accounting for kinship estimated using founder-of-origin inferred at genotyped variants as before (see file "emma_vs_gxemm.xlsx" in *Data Release: Results* for a comparison of the number of significant loci with those obtained using the EMMA model).

Our fine-mapping analysis confirmed 24 candidate loci: 5 loci unique to the additive model, 10 loci unique to the interaction model, and 9 loci significant in both models (*Table 1*). We identified body weight associations with age-dependent effects from early adulthood to adulthood: four diet-independent loci were associated with body weight exclusively during early adulthood (ages 60–160 days) and three were associated exclusively during adulthood (ages 200–660 days); the remaining seven loci were associated prior to the imposition of dietary restriction at day 180 and continued to be associated into adulthood (*Table 1*). The majority of diet-dependent loci (12 of 19) had a detectable effect on body weight at least 240 days after dietary intervention (ages 420–660 days; *Table 1*). For each candidate body weight locus, we sought to determine the likely set of causal variants and estimate the effect of the eight founder alleles in a diet- and age-dependent manner.

## Improved mapping resolution with founder allele patterns

In order to facilitate characterization and interpretation of the genetic associations at each locus, we sought to represent these associations in terms of the effects of founder haplotypes (*Zhang et al., 2022*). First, we determined the founder-of-origin for each allele at every variant that was significant in at least one age. This allowed us to assign a founder allele pattern (FAP) to each significant variant in the locus. For example, if a variant with alleles A/G had allele A in founders AJ, NZO, and PWK and the allele G in the other five founders, then we assign A to be the minor allele of this variant and define the FAP of the variant to be AJ/NZO/PWK. Next, we grouped variants based on FAP and define the logarithm of the odds (LOD) score of the FAP group to be the largest LOD score among its constituent variants. (Note that, by definition, no variant can be a member of more than one FAP group.) By focusing on the FAP groups with the largest LOD scores, we significantly reduced the number of putative causal variants (*Table 1*, see file all_significant_snps.csv in *Data Release: Results*), while representing the age- and diet-dependent effects of these loci in terms of the effects of its top FAP groups. We further narrowed the number of candidates by intersecting the variants in top FAP groups with functional annotations (e.g., gene annotations, regulatory elements, tissue-specific regulatory activity, etc.). For many loci, this procedure identified candidate regions containing 1–3 genes (*Table 1*), and we provide the full list of all genes within candidate regions in the file all_significant_genes.csv in *Data*

**Table 1.** Candidate diet-independent and diet-dependent body weight loci.

For each locus, we identified the variant with the lowest p-value at any age, the founder allele pattern (FAP) of this variant, the number of significant variants that comprise this lead FAP, the genomic positions spanning these significant FAP variants, and all ages for which at least one FAP variant is associated with body weight. For loci in which the lead FAP is comprised of fewer than 10 significant variants, we also present results for the second FAP. We list candidate genes for loci where the FAP spans three or fewer genes. Highly significant associations ($p \leq 1.1 \times 10^{-5}$) are denoted with ***, significant associations ($p \leq 10^{-4}$) with **, and suggestive associations ($p \leq 10^{-3}$) with *.

| Chrm | Founder Allele Pattern (FAP) | FAP rank | p-Value | | FAP position (Mb) Start | FAP position (Mb) End | Significant variants Total | Significant variants Open chromatin | Significant age range (days) | Age-dependent nonlinearity | Lead candidate genes (if ≤3) |
|---|---|---|---|---|---|---|---|---|---|---|---|
| Diet-independent | | | | | | | | | | | |
| | AJ/NOD | 1 | 3.24E-05 | ** | 152.046626 | 152.046626 | 1 | 1 | 60 | | Trmt1l, Edem3 |
| 1 | AJ/NOD/129 | 2 | 3.05E-05 | ** | 151.473677 | 152.280212 | 58 | 10 | 60 | 1.44E-05 | — |
| 2 | 129/CAST/PWK | 1 | 2.17E-05 | ** | 77.154962 | 77.357295 | 80 | 8 | 120–360 | 2.83E-09 | Ccdc141, Sestd1 |
| 3 | AJ/NOD/NZO/CAST | 1 | 1.89E-05 | ** | 50.533599 | 50.595073 | 55 | — | 200–260 | 5.03E-12 | Slc7a11 |
| | AJ | 1 | 6.81E-06 | *** | 58.950364 | 60.267128 | 5 | 2 | 260–360 | | Ugcg |
| 4 | PWK/WSB | 2 | 2.80E-05 | ** | 59.461248 | 59.981846 | 39 | 3 | 100–300 | 3.82E-19 | — |
| 6 | AJ/NOD | 1 | 2.75E-07 | *** | 53.61761 | 55.555977 | 89 | 2 | 60–200 | 1.55E-26 | Creb5 |
| 7 | B6/CAST | 1 | 4.20E-06 | *** | 71.375007 | 72.786849 | 47 | 1 | 80–160 | 2.48E-11 | Mctp2 |
| 7 | AJ/129/NZO/PWK | 1 | 5.70E-06 | *** | 134.08785 | 134.704465 | 20 | 2 | 80–200 | 9.17E-10 | Adam12 |
| | 129/NZO/PWK/WSB | 1 | 4.99E-06 | *** | 9.078054 | 9.078054 | 1 | — | 540–660 | | Samd5 |
| 10 | 129/NZO/WSB | 2 | 2.25E-05 | ** | 8.903387 | 9.092694 | 10 | 2 | 540 | 9.52E-06 | Sash1, Samd5 |
| 10 | CAST/PWK | 1 | 5.98E-06 | *** | 91.163191 | 91.905287 | 2,779 | 81 | 120–660 | 4.09E-10 | Anks1b, Apaf1 |
| | AJ/NZO/PWK/CAST | 1 | 5.89E-05 | ** | 58.155424 | 58.155424 | 1 | — | 80 | | — |
| 11 | B6/CAST | 2 | 4.91E-05 | ** | 56.985645 | 59.035309 | 603 | 70 | 80–100 | 2.35E-05 | — |
| 12 | NZO/CAST | 1 | 6.47E-05 | ** | 99.520559 | 99.907182 | 43 | 3 | 160–260 | 3.83E-12 | Foxn3 |
| 15 | B6/129/NZO | 1 | 4.77E-07 | *** | 99.390603 | 99.65295 | 92 | 20 | 260–600 | 1.50E-13 | Aqp2, Aqp5, Aqp6 |
| 17 | AJ/NOD/WSB | 1 | 8.56E-06 | *** | 6.753277 | 8.85311 | 149 | 11 | 60–420 | 1.23E-10 | Pde10a |
| 19 | AJ/129/NZO/PWK | 1 | 7.90E-05 | ** | 23.025043 | 23.17612 | 58 | 14 | 80–120 | 4.46E-09 | Trpm3, Klf9 |
| Diet-dependent | | | | | | | | | | | |
| 1 | NOD/CAST | 1 | 4.89E-04 | * | 151.032114 | 153.716837 | 28 | 3 | 360–660 | 3.32E-04 | — |
| 2 | AJ/CAST/PWK/WSB | 1 | 7.74E-04 | * | 22.192222 | 22.78664 | 86 | 8 | 480 | 1.32E-04 | — |
| 2 | PWK | 1 | 3.34E-04 | * | 73.590142 | 75.371564 | 1,672 | 44 | 280–300 | 4.62E-03 | — |
| 3 | NOD/CAST/PWK/WSB | 1 | 3.87E-05 | ** | 50.184746 | 50.419821 | 5 | — | 420–660 | 7.90E-04 | Slc7a11 |
| 3 | B6/PWK | 1 | 8.29E-05 | ** | 156.74554 | 156.74554 | 1 | — | 660 | 1.60E-05 | Negr1 |
| | 129/CAST/WSB | 2 | 7.33E-04 | * | 156.133466 | 156.387825 | 170 | 8 | 660 | | — |

*Table 1 continued on next page*

*Table 1 continued*

| Chrm | Founder Allele Pattern (FAP) | FAP rank | p-Value | | FAP position (Mb) | | Significant variants | | Significant age range (days) | Age-dependent nonlinearity | Lead candidate genes (if ≤3) |
|---|---|---|---|---|---|---|---|---|---|---|---|
| | | | | | Start | End | Total | Open chromatin | | | |
| | 129/NOD/PWK | 1 | 9.61E-04 | * | 57.696301 | 57.84652 | 10 | 1 | 200 | | Palm2, Pakap, Akap2 |
| 4 | NOD | 2 | 9.76E-04 | * | 57.669827 | 57.669827 | 1 | — | 200 | 5.10E-07 | — |
| 5 | NOD | 1 | 3.88E-05 | ** | 19.213904 | 21.57007 | 153 | 16 | 420–660 | 2.26E-03 | Magi2, Ptpn12 |
| 5 | PWK | 1 | 5.84E-04 | * | 68.986655 | 70.490341 | 277 | — | 360–480 | 2.28E-03 | Kctd8 |
| | AJ/129/NOD/WSB | 1 | 4.00E-05 | ** | 117.543498 | 118.050453 | 7 | — | 480–540 | | — |
| 5 | B6/CAST/PWK | 2 | 1.04E-04 | * | 116.797769 | 118.270745 | 69 | 10 | 360–600 | 1.55E-03 | — |
| 6 | B6/CAST/PWK | 1 | 1.04E-04 | * | 54.146132 | 55.452827 | 507 | 101 | 540–660 | 7.51E-04 | Ghrhr |
| 6 | B6/CAST | 1 | 1.68E-05 | ** | 139.190506 | 140.141694 | 111 | 5 | 240–420 | 9.63E-03 | Pik3c2g |
| 7 | NZO/PWK | 1 | 9.60E-04 | * | 133.838117 | 133.924023 | 218 | 3 | 420 | 3.19E-07 | Adam12 |
| | AJ/B6 | 1 | 6.29E-04 | * | 78.06102 | 79.734912 | 10 | 2 | 420–540 | | Gphn |
| 12 | WSB | 2 | 9.84E-04 | * | 78.107988 | 78.932842 | 122 | 8 | 360–420 | 4.23E-04 | Gphn |
| 12 | AJ/CAST/PWK/WSB | 1 | 3.53E-04 | * | 102.447268 | 102.56532 | 43 | 4 | 420–480 | 2.59E-05 | — |
| 13 | NZO/WSB | 1 | 1.00E-04 | ** | 117.100864 | 118.777463 | 99 | 1 | 200–300 | 1.14E-03 | Fgf10 |
| 15 | AJ/129/NOD | 1 | 3.52E-04 | * | 10.738034 | 12.040137 | 53 | 4 | 240–300 | 1.27E-04 | — |
| 17 | B6/129/NZO | 1 | 7.52E-04 | * | 6.755865 | 7.598818 | 26 | 3 | 300–360 | 3.23E-03 | — |
| 18 | 129/CAST | 1 | 3.86E-04 | * | 71.36087 | 71.58844 | 16 | 1 | 480–540 | 6.92E-03 | Dcc |
| 19 | AJ/B6/WSB | 1 | 1.24E-04 | * | 21.806665 | 22.051277 | 32 | 1 | 220–300 | 3.54E-03 | — |

*Release: Results*. In order to demonstrate the utility of this approach, we first examined a single locus on chromosome 6 strongly associated under the additive model.

We used this approach to examine a diet-independent locus on chromosome 6 significantly associated with body weight during early adulthood (*Figure 5A*). Interestingly, this same locus was nominally associated with body weight in certain dietary treatments at later ages. We hypothesized that the functional variant(s) responsible for the diet-dependent body weight association at this locus are among the variants in the respective lead FAP groups because they exhibit the strongest statistical association and it is unlikely any additional variants are segregating in this genomic interval beyond those identified in the full genome sequences of the eight founder strains. For the diet-independent locus, at age 120, we identified 87 significantly associated variants; of these, 79 could be assigned to the lead FAP group and shared a similarly high LOD score (*Figure 5B*). All of these variants are single-nucleotide polymorphisms (SNPs located in the gene *Creb5*, 78 are intronic and 1 a synonymous exon variant). For the diet-dependent locus, at age 600 days, we identified 617 variants as significant; of these, 507 could be assigned to the lead FAP and shared a similarly high LOD score (*Figure 5D*). Also, 2 of the 507 variants were intergenic structural variants; of the remaining SNPs, 5 were noncoding exon variants, 167 were intronic, and the remainder were intergenic. Given that all candidate variants were noncoding, we next sought to determine whether they were located in regulatory elements across a number of tissues, identified as regions of open chromatin measured using ATAC-seq (*Cusanovich et al., 2018*) or DNase-seq (*Gorkin et al., 2020*) (see file "chromatin_accessibility_celltypes.xlsx" in *Data Release: Results* for a description of the cell types for each of the assays, and their grouping into tissues). For the diet-independent and diet-dependent loci, we found 2 and 101 variants, respectively, that were located in regions of open chromatin (*Figure 5C and E*). Notably, both variants with diet-independent effects lay within the same muscle-specific regulatory element

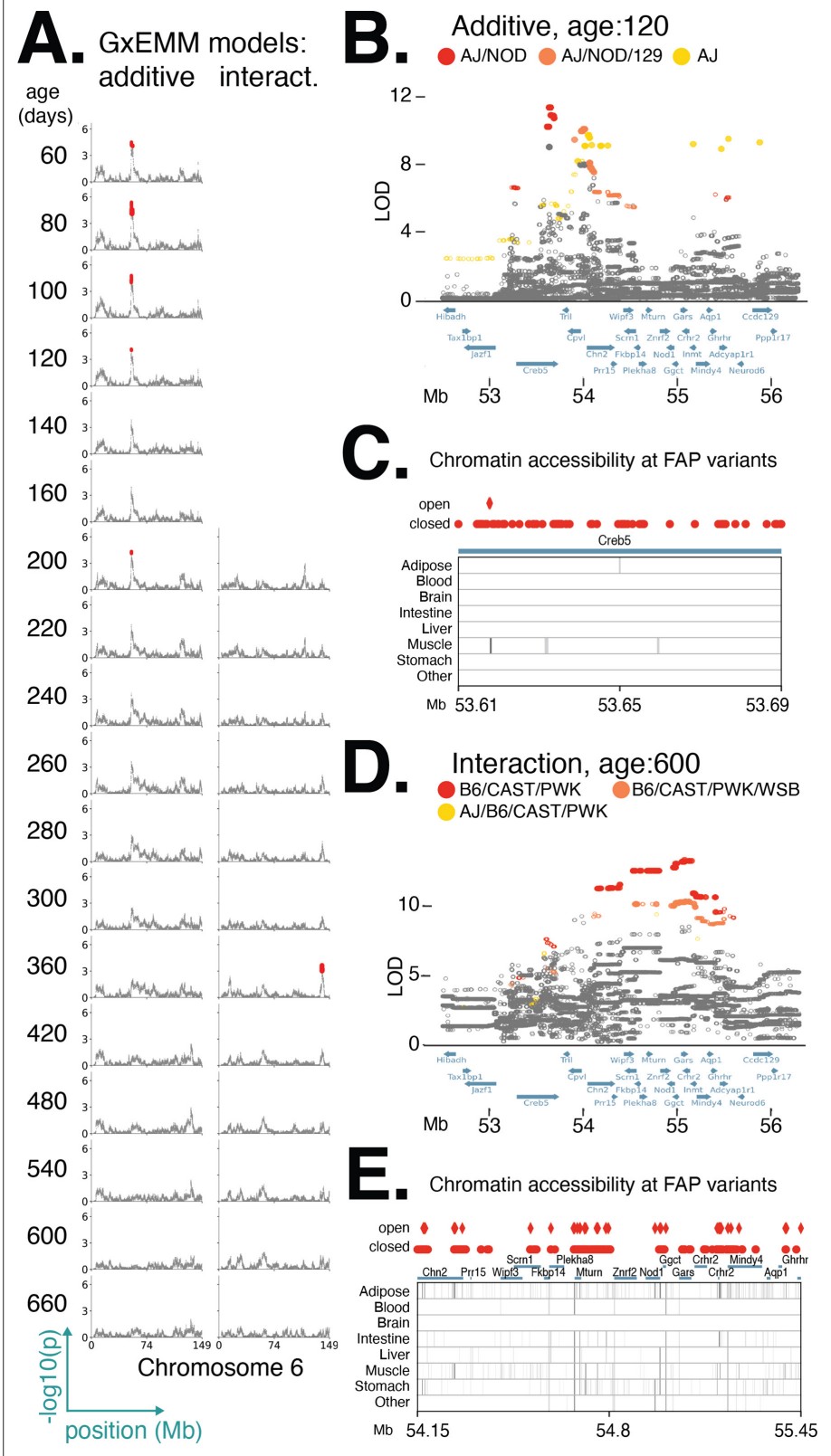

**Figure 5.** Distinct founder allele patterns (FAPs) contribute to diet-independent and diet-dependent effects on body weight within a region on chromosome 6. (**A**) Manhattan plots of additive genetic associations and genotype–diet associations on chromosome 6 at multiple ages. (**B, D**) Fine-mapping loci associated with body weight: diet-independent association at 120 days of age and diet-dependent association at 600 days of age. Solid

*Figure 5 continued on next page*

*Figure 5 continued*

circles indicate significant variants. Colors denote variants with shared FAPs; ranks 1, 2, and 3 by logarithm of the odds (LOD) score are colored red, orange, and yellow, respectively. (**C, E**) Significant variants, colored by their FAP group, along with the gene models (shown in blue) and the tissue-specific activity of regulatory elements near these variants (shown in gray). Significant variants that lie within regulatory elements are highlighted as diamonds, and regulatory elements that contain a significant variant are highlighted in black.

The online version of this article includes the following figure supplement(s) for figure 5:

**Figure supplement 1.** Age- and diet-dependent Manhattan plots for body weight, with gene–environment mixed mode (G×EMM).

**Figure supplement 2.** Age- and diet-dependent Manhattan plots for body weight, with EMMA.

**Figure supplement 3.** Loci with significant diet-independent and diet-dependent associations with body weight.

**Figure supplement 4.** Distinct founder allele patterns (FAPs) drive diet-independent and diet-dependent associations with body weight at various loci.

---

located within *Creb5*, suggesting that these variants likely affected body weight by regulating gene expression in muscle cells.

Next, we identified a locus on chromosome 12 with strong diet-dependent effects on body weight from 300 to 420 days of age (*Table 1*; *Figure 6*). Upon fine-mapping the locus with the strongest association, at 420 days of age, we identified 77 significant variants partitioning into two distinct lead FAP groups, with similarly high LOD scores and centered at the same gene (*Figure 6A and B*). The rank 1 FAP group contained variants with the minor allele specific to AJ and B6, whereas the rank 2 FAP group contained variants with the minor allele specific to WSB (*Figure 6A*). We found that the minor allele of the lead imputed variant in the AJ/B6 FAP group was associated with increased body weight in the 2D fasting diet, but had little effect on the other four diets (*Figure 6C*). In contrast, the minor allele of the lead imputed variant from the WSB-specific FAP group had the largest positive effect on body weight in the AL and 1D fast diet and largest negative effect on body weight in the 40% CR diet (*Figure 6C*). The striking difference in diet-specific effects for WSB and AJ/B6 alleles suggests that there are multiple functional variants affecting body weight at this locus (e.g., allelic heterogeneity; *Singh, 2013*). Of the 77 variants, the AJ/B6 FAP group contained six intergenic SNPs and four intronic SNPs spanning three genes: *Gphn* (gephyrin), *Plekhh1*, and *Rad51b* (*Figure 6A*). One of these 10 variants is located in a regulatory element active specifically in adipose tissue. The remaining 67 significant variants all belonged to the WSB-specific FAP group; of these, 23 SNPs were intergenic and 44 were intronic and centered at the gephyrin gene. Also, 4 of these 67 variants were located in regulatory elements active in adipose tissue as well as other tissues relevant to metabolism (*Figure 6D*).

## No evidence for pleiotropic alleles affecting diet-independent and dependent effects

We identified a diet-independent locus on chromosome 6 significantly associated with body weight during early adulthood and nominally associated with body weight in certain dietary treatments at later ages (*Figure 5A*). One explanation for this result is a single pleiotropic allele affecting body weight at two distinct stages of life: early adulthood and mid-late adulthood. Alternatively, this result could be explained by multiple alleles at a single locus (perhaps similar to the allelic heterogeneity observed in *Figure 6*) or multiple loci in tight LD. Fine-mapping this locus using the additive model, we identified the variant with the highest LOD score to be at 53.6 Mb. The minor allele at this lead variant was common to the AJ and NOD founders, while the remaining six founders possessed the alternate allele; this defined the lead diet-independent FAP at this locus to be AJ/NOD (*Figure 5B*). Separately, we fine-mapped this locus using the interaction model, identified the lead variant at 55.1 Mb, and determined the lead FAP to be B6/CAST/PWK (*Figure 5D*). These results do not support the pleiotropy hypothesis and were consistent with the hypothesis that distinct body weight alleles derived from different DO founders were responsible for the diet-dependent and diet-independent body weight associations.

We next fine-mapped other QTLs that had diet-dependent and diet-independent associations located in adjacent genomic regions to assess whether the associations were due to a single pleiotropic allele or, as we found on chromosome 6 (*Figure 5B and D*), the additive and interaction effects

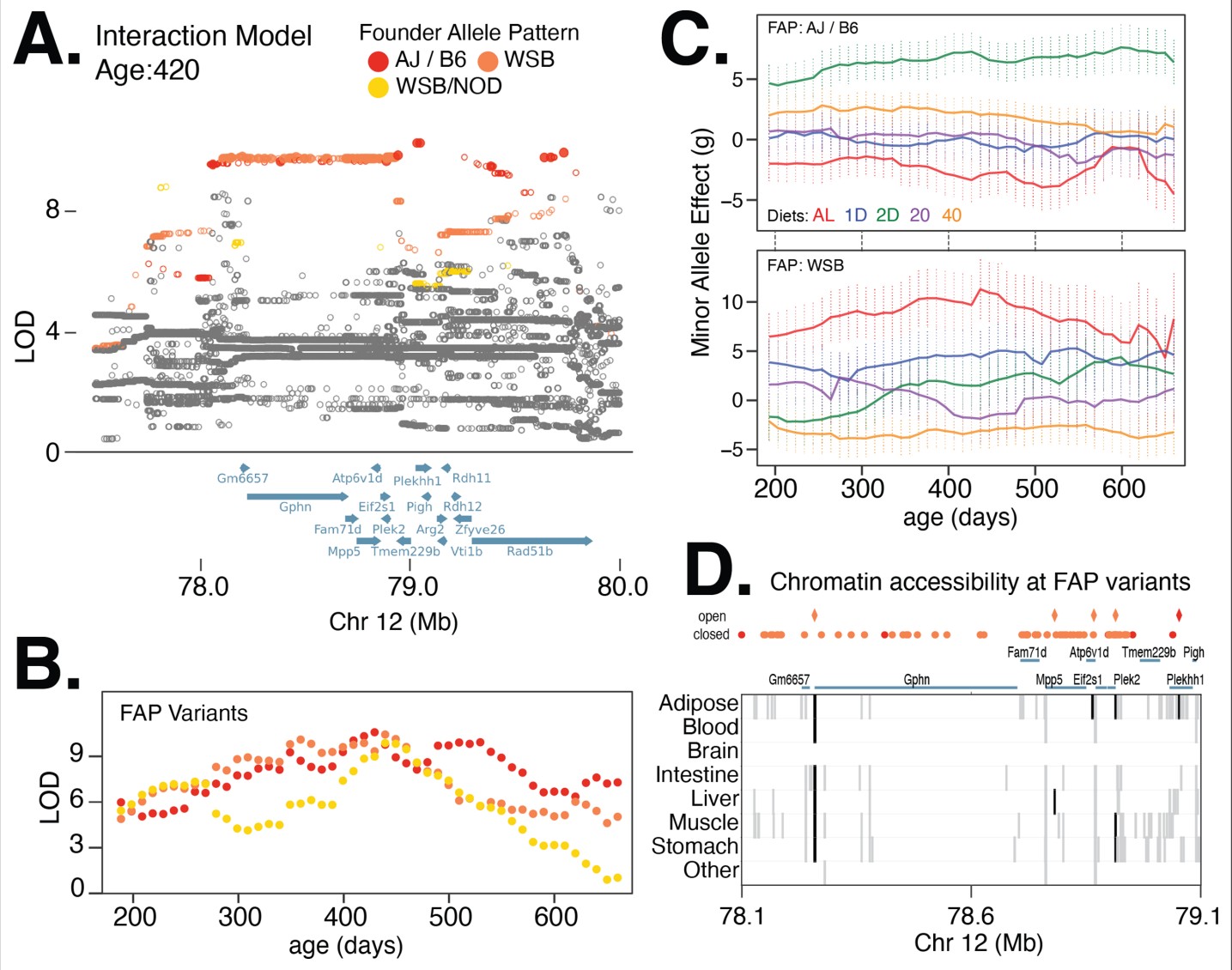

**Figure 6.** Multiple founder allele patterns (FAPs) at a body weight QTL on chromosome 12 show distinct diet-dependent effects. (**A**) Fine-mapping loci, under the interaction model, at 420 days of age. Significant variants are marked as solid circles. Colors denote variants with shared FAPs; ranks 1, 2, and 3 by logarithm of the odds (LOD) score are colored red, orange, and yellow, respectively. (**B**) Log odds ratio as a function of age, for a single variant that exhibits the strongest association from each of the top three FAPs. (**C**) Age and diet-dependent effect size of variants with the strongest associations specific to either the WSB minor allele or AJ/B6 minor allele. (**D**) Significant variants, colored by their FAP, along with the gene models (shown in blue) and the tissue-specific activity of regulatory elements near these variants (shown in gray). Significant variants that lie within regulatory elements are highlighted as diamonds and regulatory elements that contain a significant variant are highlighted in black.

were due to distinct alleles. Of the eight additional diet-dependent and diet-independent associations that colocalized to the same genomic region (*Table 1*), all of them were consistent with the hypothesis that distinct FAPs rather than a single pleiotropic allele were responsible for diet-dependent and diet-independent associations. The first six loci were located in adjacent genomic regions (*Figure 5— figure supplement 3*). The remaining two loci were composed of similar (although, not identical) FAP groups located in overlapping genomic regions (*Figure 5—figure supplement 4*). At one of these loci, on chromosome 3, the FAP driving the diet-dependent association was NOD/CAST/PWK/ WSB, whereas the diet-independent association was due to AJ/NOD/NZO/CAST (*Figure 5—figure supplement 4A*). The diet-dependent and diet-independent chromosome 7 loci are due to differential effects between NZO/PWK and AJ/129/NZO/PWK, respectively (*Figure 5—figure supplement 2D*). In summary, we identified multiple instances of diet-dependent and diet-independent associations

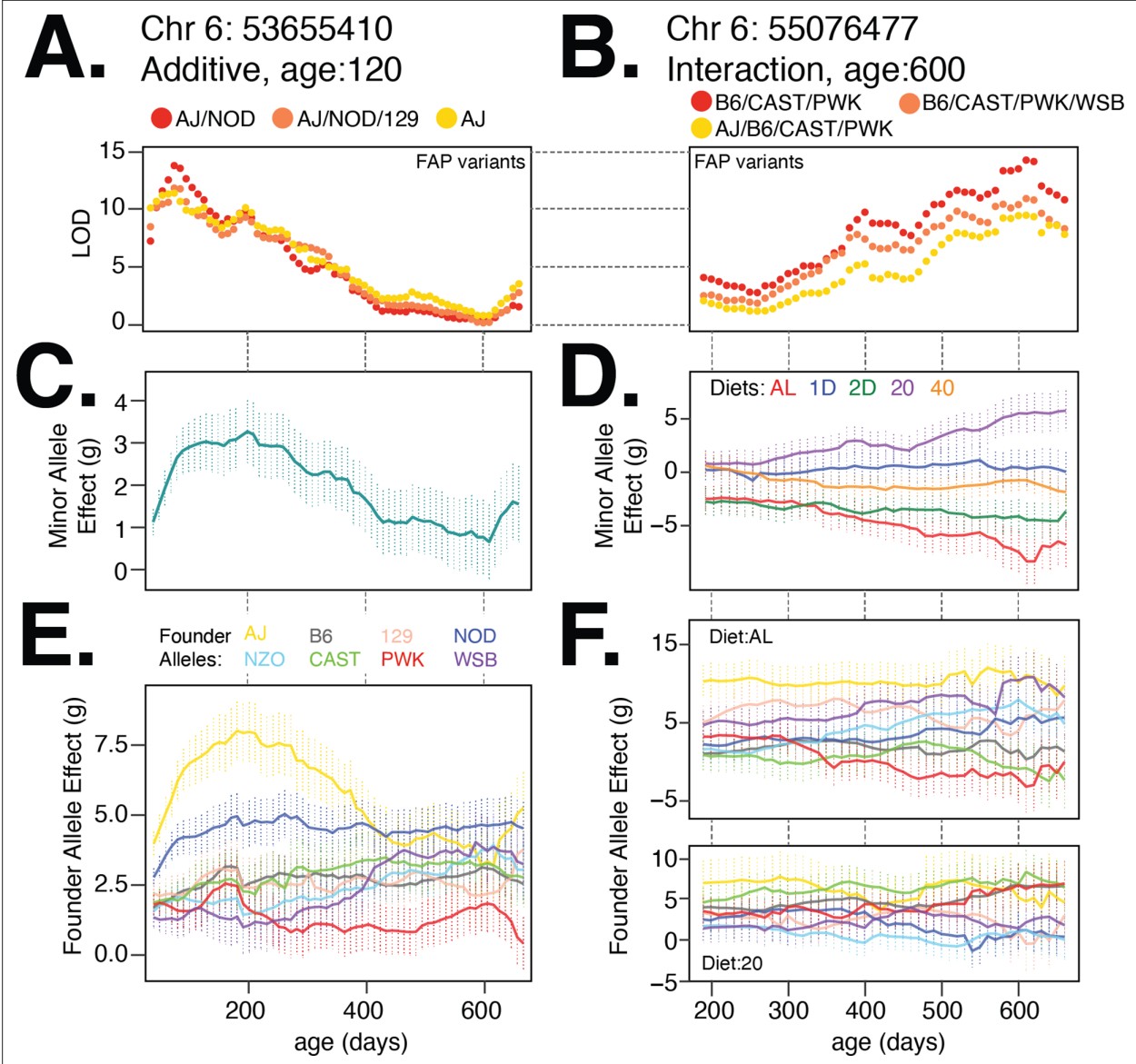

**Figure 7.** Body weight loci on chromosome 6 exhibit age- and diet-specific effects. (**A**) Log odds ratio of additive body weight association as a function of age for the lead variant from each founder allele pattern (FAP) group. Red, orange, and yellow colors denote lead variant for rank 1, 2, and 3 FAPs. (**B**) Same as (**A**) for interaction body weight association on chromosome 6. (**C, D**) Estimated mean (SE) effect on body weight (grams) of the minor allele for the lead imputed variant. For the diet-dependent locus (**D**), effects are shown for each diet treatment. (**E**) Estimated mean effect (±SE) on body weight of each founder allele for the genotyped marker with the highest logarithm of the odds (LOD) score from the additive model. (**F**) Estimated effect (±SE) on body weight of each founder allele under the ad libitum (AL) and 20% calorie restriction (CR) diets.

The online version of this article includes the following figure supplement(s) for figure 7:

**Figure supplement 1.** Diet-dependent association with body weight in a locus on chromosome 5.

**Figure supplement 2.** Nonlinear trends in genetic effects with respect to age and dietary intervention.

that mapped to the same genomic region and for all of these loci we determined that distinct FAPs were responsible for diet-dependent or diet-independent effects on body weight.

## Nonlinear changes in genetic effects with age

We found that the relationship between model LOD score and age was similar for each of the three lead FAPs at the diet-independent and diet-dependent loci on chromosome 6 (*Figure 7A and B*). The minor allele of the lead variant at the diet-independent locus was associated with increased body

weight at young ages (*Figure 7C*), whereas the minor allele for the lead diet-dependent variant had a positive effect on body weight under the 20% CR diet, a nearly neutral effect under the 40% CR, and a negative effect under the AL diet (*Figure 7D*). We next measured the effect of each founder allele at the lead diet-independent genotyped marker and, consistent with the lead FAP group for the diet-independent association, the AJ and NOD alleles had large positive effects at young ages (*Figure 7E*). For the diet-dependent association, B6, CAST, and PWK alleles were associated with decreased body weight in the AL diet and increase in body weight in the 20% CR diet (*Figure 7F*), consistent with their role in defining the lead diet-dependent FAP group. This example clearly demonstrates the insight gained by focusing on lead FAP variants to link specific founder alleles to variation in body weight and narrow the number of potential functional variants underlying body weight.

Given high temporal resolution in the measurements of body weight, we tested for nonlinearity in the trends of effect size with age for all significant loci in *Table 1*. Specifically, for each diet-independent locus, we tested for nonlinearity in the effect size trend with age in at least one founder allele of the lead variant, and for each diet-dependent locus, we tested for nonlinearity in at least one (founder, diet) pair at the lead variant (referred to as 'age-dependent nonlinearity'). In addition to diet-dependent nonlinearity, we also observed age-dependent nonlinearity at the diet-independent locus on chromosome 6 (*Figure 7C*; $p = 1.35 \times 10^{-13}$). Surprisingly, the nonlinearity at this locus appears to be largely driven by the AJ founder background (*Figure 7E*; AL, yellow trace, $p = 1.55 \times 10^{-26}$ vs. NOD, navy trace, $p = 6.5 \times 10^{-6}$), indicating that the effects of different founder alleles on body weight can have substantially different trends with age. Overall, we observed age-dependent nonlinearity to be predominant, with all 14 diet-independent loci exhibiting nonlinear trends with at least one founder allele (*Table 1*). In contrast, we observed nonlinear trends with age for diet-dependent effects to be less common, with only 4 of 19 diet-dependent loci exhibiting nonlinear trends with age in the effects of at least one founder allele under one diet. Additionally, nonlinearity was often specific to a subset of founder strains that are driving the associations at each locus, suggesting that the genetic background plays an important role when interpreting the dynamics of the genetic architecture of body weight in DO mice. Notably, such age-dependent nonlinearity often cannot be discerned even with large cross-sectional data that are typical of modern GWAS in humans.

## Nonlinear changes in diet-specific genetic effects

The diet-dependent locus on chromosome 6 illustrated a rather counterintuitive result; while we observed an approximately linear reduction in median body weight between the AL, 20% CR, and 40% CR diets in response to a linear reduction in calories (*Figure 1B*), the effects of this locus on body weight were nonlinear in the context of each diet (*Figure 7D*). We defined diet-dependent effects to have a nonlinear trend with respect to diet (referred to as 'diet-dependent nonlinearity') if the genetic effects in the context of either the 20% CR or 1D fast treatments deviated substantially from the average of the genetic effects in the context of AL and 40% CR diets or AL and 2D fast diets, respectively. We observed a second instance of diet-dependent nonlinearity at a locus we mapped to chromosome 5. This locus had the strongest diet-dependent association observed in the genome. The lead FAP group, containing variants with an allele private to the NOD strain, was associated with large positive effect on body weight in the 20% CR diet, a small positive effect in the 1D fast diet, and nearly neutral effects in the 40% CR, 2D fast, and AL diets (*Figure 7—figure supplement 1*). In total, we found this pattern to be quite common, with 9 of the 19 significant loci identified under the interaction model exhibiting diet-dependent nonlinearity (*Figure 7—figure supplement 2*).

## Candidate body weight genes linked to neurological and metabolic processes

Of the 24 significant loci from the additive and interaction models, we identified 14 loci with lead FAP groups spanning 1–3 genes (*Table 1*). Many of these genes implicated in modulating body weight are also known to affect neurological and metabolic processes.

One example is the neuronal growth regulator 1 (*Negr1*) gene, a candidate linked to body weight in mid-late adulthood via the lead 129/CAST/WSB FAP group within a diet-dependent locus on chromosome 3. This gene is highly expressed in the cerebral cortex and hippocampus in the rat brain (*Miyata et al., 2003*) and is known to regulate synapse formation of hippocampal neurons and promote neurite outgrowth of cortical neurons (*Hashimoto et al., 2008*; *Sanz et al., 2015*).

*Negr1* has also been implicated in obesity (*Lee et al., 2012*), autistic behavior, memory deficits, and increased susceptibility to seizures (*Singh et al., 2018*) in mice, and body mass index (*Speliotes et al., 2010*), and major depressive disorder (*Hyde et al., 2016*) in humans. Another example is the gephyrin gene (*Gphn*) implicated by two distinct FAP groups in the diet-dependent locus on chromosome 12 (*Figure 6A*). Gephyrin is a key structural protein at neuronal synapses that ensures proper localization of postsynaptic inhibitory receptors. Gephyrin is also known to physically interact with mTOR and is required for mTOR signaling (*Sabatini et al., 1999*), suggesting two plausible pathways for influencing body weight. On chromosome 1, murine *Trmt1l* (Trmt1-like), a gene with sequence similarity to orthologous tRNA methyltransferases in other species, was linked to body weight during early adulthood (*Figure 5—figure supplement 3A*). Mice deficient in this gene, while viable, have been found to exhibit altered motor coordination and abnormal exploratory behavior (*Vauti et al., 2007*), suggesting that the association at this locus is possibly mediated through modulating exploratory behavior. The association of these three candidates with mouse body weight is consistent with human body mass index and obesity GWAS that found an enrichment of genes active in the central nervous system (*Locke et al., 2015*; *Yengo et al., 2018*; *Schlauch et al., 2020*).

Among candidates that affect metabolic processes, *Creb5*, a gene linked to diet-independent effects on body weight (*Figure 5B*), has previously been reported to be linked to metabolic phenotypes in humans, with differential DNA methylation detected between individuals with large differences in waist circumference, hypercholesterolemia, and metabolic syndrome (*Salas-Pérez et al., 2019*). Another gene important for metabolic control in the liver, *Pik3c2g* was linked to diet-dependent effects on body weight in mid-adulthood. *Pik3c2g*-deficient mice are known to exhibit reduced liver accumulation of glycogen and develop hyperlipidemia, adiposity, and insulin resistance with age (*Braccini et al., 2015*). *Edem3*, another candidate gene at the locus on chromosome 1 linked to body weight in early adulthood (*Figure 5—figure supplement 3A*), has previously been linked to short stature in humans based on family-based exome sequencing and differential expression in chondrocytes (*Hauer et al., 2019*). Possibly sharing a similar mechanism, *Adam12*, a candidate gene in both a diet-independent and diet-dependent locus on chromosome 7, is known to play an important role in the differentiation, proliferation, and maturation of chondrocytes (*Horita et al., 2019*).

## Future considerations

To summarize, we found that the effects of age and diet on body weight differ substantially with respect to the genetic background and type of dietary intervention imposed. These results highlight that with knowledge of these environment-dependent effects, we can generate more accurate predictions of body weight trajectories than would be possible from knowledge of genotype, age, or diet alone. Moreover, with the elucidation of specific candidate genes and variants underlying these effects, we are not limited to predicting how this complex quantitative trait changes with age, but can also identify specific targets for genetic or pharmacological manipulation in an effort to improve organismal health.

One important limitation to our study is the lack of direct measurements of food consumption and feeding behaviors for each mouse in the population; this makes it difficult to ascertain how much variation in body weight can be ascribed to variation in such behaviors. Additionally, CR was imposed based on the average food consumed by a typical DO mouse rather than a per-mouse baseline of food consumption. Furthermore, CR interventions were imposed on a per-cage basis, not a per-mouse basis, because all animals were housed in groups of eight. Therefore, the social hierarchy within each cage likely contributed to additional variation in body weight (*Baud et al., 2017*). Additionally, we note that over the course of this study each animal underwent a number of phenotypic assays (*Figure 1—figure supplement 1*) that would further contribute to study-specific variation in body weight. Accounting for these sources of variation will be a promising avenue for future research, helping interpret many of the associations identified in our study.

A second important consideration for this study is the potential for survival bias to lead to inflated PVE values and false-positive associations at later ages. To evaluate the presence of survival bias, we computed PVE at all ages restricting to animals that survived to 660 days of age (75% of animals in our study). We observed similar PVE values to those from the full dataset, across the full age range, suggesting that survival bias has very little effect on the results presented in this paper (*Figure 4—figure supplement 3C*, top-left panel). However, as these animals age and the survival bias of the

population increases, genetic analyses of body weight at ages past 660 days will need to explicitly account for this effect.

In this study, we have elucidated the dynamics and context dependence of the genetic architecture of body weight from 60 to 660 days of age. By 660 days of age, nearly all surviving animals have realized their maximum adult body weight and the majority of animals have yet to experience appreciable loss of body weight indicative of late-age physiological decline. Under AL diet, reduced body weight has been associated with greater longevity, whereas under 40% CR conditions, greater longevity is associated with the maintenance of high body weight (*Liao et al., 2011*). Our future research will assess whether we observe a similar result in this DO population and determine whether alleles at body weight loci are predictive of life span in a diet-dependent manner.

## Materials and methods
### Polygenic models for gene × environment interactions

G×EMM (*Dahl et al., 2020*) is a generalization of the standard linear mixed model that allows for polygenic G×E effects. Under this model, the phenotype is written in terms of genetic effects as follows:

$$\mathbf{Y} = \alpha_0 + \sum_{e=1}^{E} \mathbf{Z}_e \alpha_e + \sum_{v=1}^{V} \mathbf{G}_v \beta_v + \sum_{e=1}^{E} \sum_{v=1}^{V} \mathbf{G}_v \mathbf{Z}_e \gamma_{ve} + \varepsilon, \tag{1}$$

where $\mathbf{Y} \in \mathbb{R}^N$ is the vector of phenotypes over $N$ samples, $\mathbf{G}_v, v = 1, \ldots, V$ are genotypes of $V$ biallelic SNPs, $\mathbf{Z}_e, e = 1, \ldots, E$ are binary vectors over $E$ environments, and $\varepsilon$ denotes the residual vector.

In our application, $\mathbf{Y}$ is the vector of $\ell_1$ trend filtered body weights at a specific age; we do not standardize the body weights so that the estimated effects are interpretable and comparable across ages. When testing for association with the founder-of-origin of markers, $\mathbf{G}_{nv} \in [0, 1]^8$; $\|\mathbf{G}_{nv}\|_1 = 1$ is a vector denoting the probability that the two alleles of the marker came from each of the eight founder lines from which the DO population is derived. $\| \cdot \|_1$ denotes the $L1$-norm of a vector. Alternatively, when testing for association with the allele dosage of a variant, $\mathbf{G}_{nv} \in [0, 2]$ is the expected allele count at variant $v$. Finally, $\mathbf{Z}_{ne} = 1$ denotes that sample $n$ is subject to environment $e$. Prior to dietary intervention, the environments in our model are the 12 generations over which the DO samples span (i.e., $E = 12$). After dietary intervention, the environments further include the five diet groups (i.e., $E = 17$).

The effects of covariates, $\boldsymbol{\alpha}$, are modeled as fixed while the genetic effects, $\boldsymbol{\beta}$, and genotype–environment effects, $\boldsymbol{\gamma}$, are modeled as random. Assuming heteroscedastic noise, $\varepsilon \sim \mathcal{N}(0, \Theta)$, a normal prior on the random genetic effects, $\beta_v \sim \mathcal{N}(0, \varrho^2/V)$, and a normal prior on the random G×E effects, $\gamma_{v\cdot} \sim \mathcal{N}(\mathbf{0}, \frac{1}{V}\Omega)$, we get $\mathbf{Y} \sim \mathcal{N}(\boldsymbol{\mu}, \Lambda)$, where $\boldsymbol{\mu} = \alpha_0 + \sum_e \mathbf{Z}_e \alpha_e$ and $\Lambda = \Theta + \varrho^2 \mathcal{K} + \sum_{e,e'} \Omega_{ee'} \left( \mathcal{K} \circ \left( \mathbf{Z}_e \mathbf{Z}_{e'}^T \right) \right)$ is a diagonal matrix with entries specified as $\Theta_{nn} = \sum_e \mathbf{Z}_{ne}^2 \sigma_e^2$, $\mathcal{K}$ is the kinship matrix with entries defined as $\mathcal{K}_{mn} = \frac{1}{V} \sum_v G_{mv}^T G_{nv}$, $\varrho^2$ is the variance of environment-independent genetic effects, $\Omega \in \mathbb{R}^{E \times E}$ is the variance–covariance matrix representing the co-variation in environment-dependent genetic effects between pairs of environments, and $A \circ B$ denotes the Hadamard product of matrices $A$ and $B$. We do not include an environment-independent noise term since it is nonidentifiable.

Off-diagonal elements of $\Omega$ quantify the covariance in the interaction effect sizes between pairs of environments; thus, a diagonal $\Omega$ assumes that the interaction effect sizes between pairs of environments are uncorrelated. This assumption of diagonal $\Omega$ could lead to some bias in the estimates of the variance components, although bias in PVE estimates are expected to be minimal. We computed PVE estimates using a full $\Omega$ and a diagonal $\Omega$ and observed little difference (*Figure 4—figure supplement 5*). For the sake of computational simplicity, we assumed all off-diagonal elements of $\Omega$ to be zero in this study. Note that each of the above parameters and data vectors in the model may be distinct at different ages.

Recent work has shown that non-negativity constraints on the variance components are both important in order to obtain unbiased estimates and unnecessary to achieve valid estimates of heritability (*Steinsaltz et al., 2020*). Nevertheless, we constrained the variance components to be non-negative since they are more interpretable in the context of our study. While it is likely that this constraint contributes to the small bias observed in our simulations (*Figure 3*), we observed almost no bias in heritability estimates for body weight due to the constraint, likely because of strong signals

in the data (*Figure 4—figure supplement 4*). We have also included a flag in our published code to relax this constraint when estimating the variance components.

## Proportion of phenotypic variance explained by genetics

Decomposing the phenotype into genetic and nongenetic effects, $\mathbf{Y} = \mathbf{Y_G} + \mathbf{Y_\varepsilon}$, the expected proportion of phenotypic variance explained by genetic effects is approximately given as

$$\text{PVE} = \mathbb{E}\left[\frac{\mathbb{V}[\mathbf{Y_G}]}{\mathbb{V}[\mathbf{Y}]}\right] \approx \frac{\mathbb{E}[\mathbb{V}[\mathbf{Y_G}]]}{\mathbb{E}[\mathbb{V}[\mathbf{Y}]]} \frac{\mathbf{Var}_G}{\mathbf{Var}_Y}, \tag{2}$$

where $\mathbb{V}[\cdot]$ denotes the sample variance. The expected sample phenotypic variance conditional on environment $e$ can be written as

$$\mathbb{E}\left[\mathbb{V}[\mathbf{Y}|e]\right] = \frac{\mathbb{E}\left[\sum_n Y_n^2 Z_{ne}\right]}{N_e} - \frac{\mathbb{E}\left[\left(\sum_n Y_n Z_{ne}\right)^2\right]}{N_e^2}, \tag{3}$$

where $Z_{ne}$ is an indicator variable denoting whether sample $n$ belongs to environment $e$, and $N_e = \sum_n Z_{ne}$ is the number of samples in environment $e$. Under the G×EMM model, starting from *Equation 1* and integrating out the random effects, we can write the numerator of the first term in the expectation as

$$\mathbb{E}\left[\sum_n Y_n^2 Z_{ne}\right] = \sum_n \mu_n^2 Z_{ne} + \frac{\varrho^2}{V}\sum_{n,v} G_{nv}^2 Z_{ne} \\ + \sum_{e',e''}\frac{\Omega_{e'e''}}{V}\sum_{n,v}G_{nv}^2 Z_{ne} Z_{ne'} Z_{ne''} + \sum_n \Theta_{nn}Z_{ne}, \tag{4}$$

and the numerator of the second term in the expectation as

$$\mathbb{E}\left[\left(\sum_n Y_n Z_{ne}\right)^2\right] = \left(\sum_n \mu_n Z_{ne}\right)^2 + \frac{\varrho^2}{V}\sum_v \sum_{n,n'} G_{nv}G_{n'v}Z_{ne}Z_{n'e} \\ + \sum_{e',e''}\frac{\Omega_{e'e''}}{V}\sum_v \sum_{n,n'} G_{nv}G_{n'v}Z_{n,e'}Z_{n'e''}Z_{ne}Z_{n'e} \\ + \sum_n \Theta_{nn}Z_{ne}, \tag{5}$$

where $e'$ and $e''$ iterate over all environments and $n'$ and $n''$ iterate over all samples. Therefore, the expected sample phenotypic variance can be decomposed as follows:

$$\mathbb{E}\left[\mathbb{V}[\mathbf{Y}|e]\right] = \left(\frac{\sum_n \mu_n^2 \mathbf{Z}_{ne}}{N_e} - \frac{(\sum_n \mu_n \mathbf{Z}_{ne})^2}{N_e^2}\right) \\ + \varrho^2\left(\frac{\mathbf{tr}(\mathcal{K}\circ W_e)}{N_e} - \frac{\mathbf{sum}(\mathcal{K}\circ W_e)}{N_e^2}\right) \\ + \sum_{e'}\Omega_{e'e'}\left(\frac{\mathbf{tr}(\mathcal{K}\circ W_e \circ W_{e'})}{N_e} - \frac{\mathbf{sum}(\mathcal{K}\circ W_e \circ W_{e'})}{N_e^2}\right) \\ + \mathbf{tr}(\Theta \circ W_e)\left(\frac{N_e - 1}{N_e^2}\right), \tag{6}$$

where $W_e = \mathbf{Z}_e\mathbf{Z}_e^T$, $\mathbf{tr}(\cdot)$ denotes that trace of a matrix, $\mathbf{sum}(\cdot)$ denotes the sum of all entries of the matrix, and $A \circ B$ denotes the Hadamard product of matrices $A$ and $B$. The first term quantifies the phenotypic variance explained by fixed effects, the second and third terms together quantify the phenotypic variance explained by genetic effects ($\mathbb{E}\left[\mathbb{V}[\mathbf{Y_G}|e]\right]$), and the fourth term quantifies the residual (unexplained) phenotypic variance. The total proportion of variance explained by genetics in the entire sample can now be computed using *Equation 2*.

$$\text{PVE}_{tot} = \frac{\mathbf{Var}_G}{\mathbf{Var}_Y}, \tag{7}$$

$$\mathbf{Var}_G = \varrho^2\left(\frac{\mathbf{tr}(\mathcal{K})}{N} - \frac{\mathbf{sum}(\mathcal{K})}{N^2}\right) \\ + \sum_e \Omega_{ee}\left(\frac{\mathbf{tr}(\mathcal{K}\circ W_e)}{N} - \frac{\mathbf{sum}(\mathcal{K}\circ W_e)}{N^2}\right), \tag{8}$$

$$\mathbf{Var}_Y = \mathbf{Var}_G$$

$$+ \left( \frac{\left( \sum_n \mu_n^2 \right)}{N} - \frac{\left( \sum_n \mu_n \right)^2}{N^2} \right) \tag{9}$$

$$+ \mathbf{tr}(\Theta) \left( \frac{N-1}{N^2} \right).$$

The two terms in $\mathbf{Var}_G$ are genetic contributions to phenotypic variation that are shared across environments and specific to environments, respectively. The second and third terms in $\mathbf{Var}_Y$ are phenotypic variation explained by fixed effects and unexplained residual phenotypic variation, respectively.

The expected total sample phenotypic variance (across all environments) again has two terms, as in *Equation 3*; the numerator of the first term is written as

$$\mathbb{E}\left[ \sum_n Y_n^2 \right] = \sum_n \mu_n^2 + \frac{\varrho^2}{V} \sum_n \sum_v G_{nv}^2$$
$$+ \sum_{e,e'} \frac{\Omega_{ee'}}{V} \sum_n \sum_v G_{nv}^2 Z_{ne} Z_{ne'} + \sum_n \Theta_{nn}, \tag{10}$$

and the numerator of the second term is written as

$$\mathbb{E}\left[ \left( \sum_n Y_n \right)^2 \right] = \left( \sum_n \mu_n \right)^2 + \frac{\varrho^2}{V} \sum_v \sum_{n,n'} G_{nv} G_{n'v}$$
$$+ \sum_{e,e'} \frac{\Omega_{ee'}}{V} \sum_v \sum_{n,n'} G_{nv} G_{n'v} Z_{ne} Z_{n'e'} + \sum_n \Theta_{nn}. \tag{11}$$

The expected total sample phenotypic variance can be decomposed as follows:

$$\mathbb{E}\left[ \mathbb{V}\left[ Y \right] \right] = \left( \frac{\sum_n \mu_n^2}{N} - \frac{\left( \sum_n \mu_n \right)^2}{N^2} \right)$$
$$+ \varrho^2 \left( \frac{\mathbf{tr}(\mathcal{K})}{N} - \frac{\mathbf{sum}(\mathcal{K})}{N^2} \right)$$
$$+ \sum_e \Omega_{ee} \left( \frac{\mathbf{tr}(\mathcal{K} \circ W_e)}{N} - \frac{\mathbf{sum}(\mathcal{K} \circ W_e)}{N^2} \right)$$
$$+ \mathbf{tr}(\Theta) \left( \frac{N-1}{N^2} \right). \tag{12}$$

The proportion of variance explained by genetics conditional on environment can be computed by substituting the above in *Equation 2*.

$$\mathrm{PVE}_e = \frac{\mathbf{Var}_{G|e}}{\mathbf{Var}_{Y|e}}, \tag{13}$$

$$\mathbf{Var}_{G|e} = \varrho^2 \left( \frac{\mathbf{tr}(\mathcal{K} \circ W_e)}{N_e} - \frac{\mathbf{sum}(\mathcal{K} \circ W_e)}{N_e^2} \right)$$
$$+ \sum_{e'} \Omega_{e'e'} \left( \frac{\mathbf{tr}(\mathcal{K} \circ W_e \circ W_{e'})}{N_e} - \frac{\mathbf{sum}(\mathcal{K} \circ W_e \circ W_{e'})}{N_e^2} \right), \tag{14}$$

$$\mathbf{Var}_{Y|e} = \mathbf{Var}_{G|e}$$

$$+ \left( \frac{\left( \sum_n \mu_n^2 \mathbf{Z}_{ne} \right)}{N_e} - \frac{\left( \sum_n \mu_n \mathbf{Z}_{ne} \right)^2}{N_e^2} \right) \tag{15}$$

$$+ \mathbf{tr}(\Theta \circ W_e) \left( \frac{N_e - 1}{N_e^2} \right).$$

The proportion of phenotypic variance explained by genetic effects is equivalent to narrow-sense heritability, once variation due to additive effects of environment, batch, and other study design artifacts have been removed. In this work, we use the more general term, proportion of variance explained, to accommodate variation due to effects of diet and environment.

Under the EMMA model, the expected total sample phenotypic variance simplifies to

$$\mathbb{E}\left[ \mathbb{V}\left[ Y \right] \right] = \left( \frac{\sum_n \mu_n^2}{N} - \frac{\left( \sum_n \mu_n \right)^2}{N^2} \right)$$
$$+ \varrho^2 \left( \frac{\mathbf{tr}(\mathcal{K})}{N} - \frac{\mathbf{sum}(\mathcal{K})}{N^2} \right)$$
$$+ \theta^2 \left( \frac{N-1}{N^2} \right), \tag{16}$$

where $\theta$ denotes the homoscedastic noise. The first component quantifies the phenotypic variance explained by fixed effects, the second component quantifies the phenotypic variance explained by genetic effects ($\mathbb{E}\left[\mathbb{V}\left[\mathbf{Y_G}\right]\right]$), and the third component quantifies the residual (unexplained) phenotypic variance. Substituting these into *Equation 2* gives us the proportion of variance explained by genetics under the EMMA model.

## Genome-wide association mapping

### Additive genetic effects

To test for additive effect of a genetic variant on the phenotype, we include the focal variant among the fixed effects in the model while treating all other variants to have random effects.

$$\mathbf{Y} = \sum_c \mathbf{X}_c \alpha_c + \phi_s \mathbf{G}_s + \sum_v \mathbf{G}_v \beta_v + \sum_{v,e} \mathbf{G}_v \mathbf{Z}_e \gamma_{ve} + \varepsilon \tag{17}$$

Applying the priors described above for $\beta_v$, $\boldsymbol{\gamma_v}$, and $\varepsilon$, we can derive the corresponding mixed-effects model is as follows:

$$\mathbf{Y} \sim \mathcal{N}\left(\sum_c \mathbf{X}_c \alpha_c + \phi_s \mathbf{G}_s, \Lambda_s\right), \tag{18}$$

where $\Lambda_s = \Theta + \varrho^2 \mathcal{K}_s + \sum_e \Omega_{ee}\left(\mathcal{K}_s \circ W_e\right)$ and $\mathcal{K}_s$ is the kinship matrix after excluding the entire chromosome containing the variant $s$ (leave-one-chromosome-out [LOCO] kinship). Leaving out the focal chromosome when computing kinship increases our power to detect associations at the focal variant (*Lippert et al., 2011*). The test statistic is the log likelihood ratio $\Phi^a(\mathbf{Y}, \mathbf{G}_s)$ comparing the alternate model $\mathcal{H} : \phi_s \neq 0$ to the null model $\mathcal{H}_0 : \phi_s = 0$.

$$\Phi^a(\mathbf{Y}, \mathbf{G}_s) = \log \frac{\max \mathcal{L}(\phi_s, \boldsymbol{\alpha}, \boldsymbol{\sigma}, \varrho, \Omega)}{\max \mathcal{L}(\phi_s=0, \boldsymbol{\alpha}, \boldsymbol{\sigma}, \varrho, \Omega)} \tag{19}$$

### Genotype–environment effects

To test for effects of interaction between genotype and environment on the phenotype, we include a fixed effect for the focal variant and its interactions with the set of all environments of interest ($\mathcal{E}$) while treating all other variants to have random effects for their additive and interaction contributions:

$$\mathbf{Y} = \sum_c \mathbf{X}_c \alpha_c + \phi_s \mathbf{G}_s + \sum_{e \in \mathcal{E}} \chi_{se} \mathbf{G}_s \mathbf{Z}_e + \sum_{v \neq s} \mathbf{G}_v \beta_v + \sum_{v \neq s,e} \mathbf{G}_v \mathbf{Z}_e \gamma_{ve} + \varepsilon. \tag{20}$$

The corresponding mixed-effects model is

$$\mathbf{Y} \sim \mathcal{N}\left(\sum_c \alpha_c \mathbf{X}_c + \phi_s \mathbf{G}_s + \sum_{e \in \mathcal{E}} \chi_{se} \mathbf{G}_s \mathbf{Z}_e, \Lambda_s\right), \tag{21}$$

and the test statistic is:

$$\Phi^i(\mathbf{Y}, \mathbf{G}_s) = \log \frac{\max \mathcal{L}(\chi_{s.}, \phi_s, \boldsymbol{\alpha}, \boldsymbol{\sigma}, \varrho, \Omega)}{\max \mathcal{L}(\chi_{s.}=0, \phi_s, \boldsymbol{\alpha}, \boldsymbol{\sigma}, \varrho, \Omega)}, \tag{22}$$

where $\chi_{s.}$ is a vector notation for all $\chi_{se}$. Note that this statistic tests for the presence of interaction effects between the focal variant and any of the environments of interest.

## Likelihood and gradients for G×EMM model

Under the full G×EMM model, the phenotype $\mathbf{Y}$ depends on fixed and random effects as follows:

$$\mathbf{Y} \sim \mathcal{N}(\boldsymbol{\mu}, \Lambda), \tag{23}$$

where $\boldsymbol{\mu} = \alpha_0 + \sum_e \mathbf{Z}_e \alpha_e$ captures all fixed effects and $\Lambda = \Theta + \varrho^2 \mathcal{K} + \sum_e \Omega_{ee} \mathcal{K} \circ \left(\mathbf{Z}_e \mathbf{Z}_e^T\right)$ captures the covariance after integrating out the random effects. The parameters to be estimated in this model are $\boldsymbol{\alpha}$, $\boldsymbol{\sigma}$, $\varrho$, and $\Omega$. The complete log likelihood can be written as

$$\mathcal{L} = -\frac{N}{2} \log(2\pi) - \frac{1}{2} \log \det \Lambda - \frac{1}{2} \left(Y - X\boldsymbol{\alpha}\right)^T \Lambda^{-1} \left(Y - X\boldsymbol{\alpha}\right). \tag{24}$$

Maximizing the log likelihood over $\boldsymbol{\alpha}$ gives us

$$\hat{\boldsymbol{\alpha}} = \arg\max_{\boldsymbol{\alpha}} \mathcal{L} = \left(X^T \Lambda^{-1} X\right)^{-1} \left(X^T \Lambda^{-1} Y\right). \tag{25}$$

Substituting this into the log likelihood, we get

$$\mathcal{L}_{\hat{\alpha}} = -\frac{N}{2}\log(2\pi) - \frac{1}{2}\log\det\Lambda - \frac{1}{2}Y^TPY, \tag{26}$$

where $P = \Lambda^{-1} - \Lambda^{-1}X\left(X^T\Lambda^{-1}X\right)^{-1}X^T\Lambda^{-1}$.

Computing the gradient of $\mathcal{L}_{\hat{\alpha}}$ involves evaluating the following gradients:

$$\frac{\partial\Lambda}{\partial\sigma_e^2} = \mathcal{I}_{Z_e} \tag{27}$$

$$\frac{\partial\Lambda}{\partial\varrho^2} = \mathcal{K} \tag{28}$$

$$\frac{\partial\Lambda}{\partial\Omega_{ee}} = \mathcal{K}\circ(Z_eZ_e^T) \tag{29}$$

$$\partial\log\det\Lambda = \mathbf{tr}\left(\Lambda^{-1}\partial\Lambda\right) \tag{30}$$

$$\partial\Lambda^{-1} = -\Lambda^{-1}\partial\Lambda\Lambda^{-1} \tag{31}$$

$$\partial P = -P\partial\Lambda P, \tag{32}$$

where $\mathcal{I}_{\mathbf{Z}_e}$ is a diagonal matrix with elements of the vector $\mathbf{Z}_e$ on the diagonal. Thus,

$$\frac{\partial\mathcal{L}_{\hat{\alpha}}}{\partial\sigma_e^2} = -\frac{1}{2}\mathbf{tr}(\Lambda^{-1}\mathcal{I}_{Z_e}) + \frac{1}{2}Y^TP\mathcal{I}_{Z_e}PY \tag{33}$$

$$\frac{\partial\mathcal{L}_{\hat{\alpha}}}{\partial\varrho^2} = -\frac{1}{2}\mathbf{tr}\left(\Lambda^{-1}\mathcal{K}\right) + \frac{1}{2}Y^TP\mathcal{K}PY \tag{34}$$

$$\frac{\partial\mathcal{L}_{\hat{\alpha}}}{\partial\Omega_{ee}} = -\frac{1}{2}\mathbf{tr}\left(\Lambda^{-1}\mathcal{K}^e\right) + \frac{1}{2}Y^TP\mathcal{K}^ePY, \tag{35}$$

where $\mathcal{K}^e = \mathcal{K}\circ(Z_eZ_e^T)$.

## Permutations

Since the assumptions underlying the G×EMM model are unlikely to hold exactly in practice, we used permutations to assess the significance of the observed values of $\Phi^a$ and $\Phi^i$. Specifically, for each variant $s$, we generated $M$ independent random permutations $\mu_1,\ldots,\mu_M$ of $(1,\ldots,N)$ and computed test statistics $\Phi_m^{a/i}\Phi^{a/i}\left(\mu_m(\mathbf{Y}),\mathbf{G}_s\right)$ for each permutation $\mu_m$. The p-value associated with $\Phi^a$ and $\Phi^i$ is then given as

$$p^{a/i} = \frac{\#\{m:\Phi_m^{a/i}\geq\Phi^{a/i}\}+1}{M+1} \tag{36}$$

We used a sequential procedure (*Besag and Clifford, 1991*; *Shim and Stephens, 2015*) to compute the p-value that helps avoid large numbers of permutations for nonsignificant results, substantially speeding up computation. Specifically, instead of a fixed large number of permutations $M$, we performed $M_s$ permutations for each variant $s$ until $C_s^{a/i}\#\{m:\Lambda_m^{a/i}\geq\Lambda^{a/i}\} = 10$ or $M_s = 10^8$. The p-value for the $s$th variant is then a random sample from the interval $[\frac{C_s}{M_s},\frac{C_s+1}{M_s+1}]$. This scheme allows us to estimate p-value quickly without carrying out large numbers of permutations at every marker. Although the precision of this estimate is lower for markers that are not significant (i.e., most of the genome), precision increases as the significance of association increases.

## Tests for nonlinearity in trends of effect size with age

We test for nonlinear dependence between effect size and age by comparing two models: a null model that only allows for linear dependence between age and effect size and an alternate model that allows for dependence between effect size and higher-order powers of age. While this test can be applied to evaluate nonlinear effect-size trends for any covariate (e.g., dietary intervention, genotype, etc.), we focus our attention to tests for nonlinear trends in the effect sizes of the founder alleles and the effect sizes of the interaction between founder alleles and diet.

Specifically, the estimated effects $\hat{\phi}_{st}$ and their standard errors $\hat{\psi}_{st}$ can be modeled as

$$\hat{\phi}_{st} \sim \mathcal{N}(\phi_{st},\hat{\psi}_{st}); \ t\in\{40,50,\ldots,660\} \tag{37}$$

and the two models of the effect size trends are

$$\mathcal{H}_{n\ell} \quad : \phi_{st} = \varphi_0 + \varphi_1 t + \varphi_2 t^2 + \varphi_3 t^3$$
$$\mathcal{H}_{\ell} \quad : \phi_{st} = \varphi_0 + \varphi_1 t, \tag{38}$$

where $t$ denotes the ages (in days) at which we test for association between a genetic variant and body weight and $\phi_{st}$ the unobserved effect sizes at these ages. We used a likelihood ratio test statistic ($\aleph$) to compare the two models and quantify the evidence for nonlinearity in effect size trends with age and assessed significance using p-values computed using a chi-squared distribution with degree-of-freedom 2.

$$\aleph = 2 \ln \frac{\max \mathcal{L}(\varphi)}{\max \mathcal{L}(\varphi; \varphi_2 = \varphi_3 = 0)}, \tag{39}$$

where $\mathcal{L}(\varphi)$ is the likelihood under the model in *Equation 37*.

We apply this test for each of the founder effects in each of the 14 diet-independent loci (112 tests) and each of the 19 founder-x-diet interaction effects in each of the diet-dependent loci (760 tests). We assign a diet-independent trend as significantly nonlinear if we observe a $p < 10^{-4}$ and assign a diet-dependent trend as significantly nonlinear if we observe a $p < 10^{-5}$ (Bonferroni-corrected p-value thresholds).

## Data and software release

Data used in this study, and significant QTLs, can be downloaded from the following links:

- Genotypes: https://doi.org/10.6084/m9.figshare.13190735
- Body Weight: https://doi.org/10.6084/m9.figshare.13190702
- Covariates: https://doi.org/10.6084/m9.figshare.13190615
- Results (QTLs, functional annotations): https://doi.org/10.6084/m9.figshare.13190708

Our Python implementation of the EMMA and G×EMM models can be downloaded from https://github.com/calico/do_qtl; (*Raj, 2021*; copy archived at swh:1:rev:3d23d3fbd7768b74b0bba6183de1570a65a2d93d).

## Acknowledgements

The authors acknowledge Madeleine Cule, Anastasia Baryshnikova, Dale Zhang, and Nicholas Bernstein for their comments on the article and reviewing the software developed for all analyses, and Adam Baker for helping with the design of *Table 1*. The authors would also like to acknowledge Amelie Baud, Andrew Dahl, and an anonymous reviewer for their insightful comments and suggestions that helped greatly improve our article.

# Additional information

## Competing interests

Kevin M Wright: Kevin is an employee of Calico Life Sciences LLC. Andrea Di Francesco: Andrea is an employee of Calico Life Sciences LLC. Adam Freund: Adam is an employee of Calico Life Sciences LLC. Vladimir Jojic: Vladimir is an employee of Calico Life Sciences LLC. Anil Raj: Anil is an employee of Calico Life Sciences LLC. The other authors declare that no competing interests exist.

## Funding

| Funder | Grant reference number | Author |
|---|---|---|
| Calico Life Sciences LLC | | Kevin M Wright<br>Andrew G Deighan<br>Andrea Di Francesco<br>Adam Freund<br>Vladimir Jojic<br>Gary A Churchill<br>Anil Raj |

| Funder | Grant reference number | Author |
|--------|------------------------|--------|

The funders had no role in study design, data collection and interpretation, or the decision to submit the work for publication.

## Author contributions

Kevin M Wright, Conceptualization, Formal analysis, Investigation, Methodology, Validation, Visualization, Writing – original draft; Andrew G Deighan, Data curation, Resources; Andrea Di Francesco, Validation, Writing - review and editing; Adam Freund, Funding acquisition, Project administration, Supervision; Vladimir Jojic, Methodology, Software, Writing - review and editing; Gary A Churchill, Conceptualization, Funding acquisition, Project administration, Resources, Writing - review and editing; Anil Raj, Conceptualization, Data curation, Formal analysis, Investigation, Methodology, Project administration, Software, Supervision, Validation, Visualization, Writing – original draft

## Author ORCIDs

Kevin M Wright ⓘ http://orcid.org/0000-0001-8772-2687
Andrea Di Francesco ⓘ http://orcid.org/0000-0001-6867-8203
Adam Freund ⓘ http://orcid.org/0000-0002-7956-5332
Gary A Churchill ⓘ http://orcid.org/0000-0001-9190-9284
Anil Raj ⓘ http://orcid.org/0000-0003-4412-0883

## Ethics

All animal procedures were reviewed and approved by the Jackson Laboratory Animal Care and Use Committee (summary # 06005).

## Decision letter and Author response

Decision letter https://doi.org/10.7554/eLife.64329.sa1
Author response https://doi.org/10.7554/eLife.64329.sa2

# Additional files

## Supplementary files

• Transparent reporting form

## Data availability

All data used in the study, and results of analyses, are publicly available at https://figshare.com/projects/Age_and_diet_shape_the_genetic_architecture_of_body_weight_in_Diversity_Outbred_mice/92201.

The following datasets were generated:

| Author(s) | Year | Dataset title | Dataset URL | Database and Identifier |
|-----------|------|---------------|-------------|-------------------------|
| Wright KM, Deighan A, Di Francesco A, Freund A, Jojic V, Churchill G, Raj A | 2020 | Genotype data: Age and diet shape the genetic architecture of body weight in Diversity Outbred mice | https://figshare.com/articles/dataset/Genotype_data/13190735 | figshare, 10.6084/m9.figshare.13190735 |
| Wright KM, Deighan A, Di Francesco A, Freund A, Jojic V, Churchill G, Raj A | 2020 | Phenotype data: Age and diet shape the genetic architecture of body weight in Diversity Outbred mice | https://figshare.com/articles/dataset/Phenotype_data/13190702 | figshare, 10.6084/m9.figshare.13190702 |
| Wright KM, Deighan A, Di Francesco A, Freund A, Jojic V, Churchill G, Raj A | 2020 | Results: Age and diet shape the genetic architecture of body weight in Diversity Outbred mice | https://figshare.com/articles/dataset/Significant_variants_genes/13190708 | figshare, 10.6084/m9.figshare.13190708 |
| Wright KM, Deighan A, Di Francesco A, Freund A, Jojic V, Churchill G, Raj A | 2020 | Covariate data: Age and diet shape the genetic architecture of body weight in Diversity Outbred mice | https://figshare.com/articles/dataset/Covariate_data/13190615 | figshare, 10.6084/m9.figshare.13190615 |

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
