## [Editor Report]

This is an outstanding dissection of the genetic architecture of body weight at the genome-wide level across time and across environments. The use of a multiparental mouse population permits high-resolution mapping. The statistical analyses are advanced, leveraging new models, as well as tools developed specifically for this mouse population. The corresponding results are presented in nice and informative figures.

---

## [Decision Letter]

**Decision letter after peer review:**

Thank you for submitting your article "Age and diet shape the genetic architecture of body weight in Diversity Outbred mice" for consideration by *eLife*. Your article has been reviewed by 3 peer reviewers, and the evaluation has been overseen by a Reviewing Editor and Matt Kaeberlein as the Senior Editor. The following individuals involved in review of your submission have agreed to reveal their identity: Andrew Dahl (Reviewer #2); Amelie Baud (Reviewer #3).

The reviewers have discussed the reviews with one another and the Reviewing Editor has drafted this decision to help you prepare a revised submission.

Summary:

Wright et al., have empirically studied the subtleties of the genetic architecture of a classic quantitative trait, body weight, providing evidence of gene-by-age and gene-by-diet effects, both at the level of the entire genome and at the level of individual genetic loci. The authors also identified likely causal variants at individual loci. To do so, they used a mouse population descended from multiple, known founders and reconstructed the founder haplotypes at each locus, and also used genomic annotations. Finally, the authors also explored pleiotropy, allelic heterogeneity and locus heterogeneity at the associated loci.

This is an outstanding dissection of the genetic architecture of a complex trait at the genome-wide level, at the locus level, across time and across environments. The experimental design is excellent, with the use of a multiparental mouse population that also permits high-resolution mapping. The statistical analyses are good and advanced, leveraging the GxEMM model recently published by Dahl et al., as well as tools developed specifically for this mouse population to trace back the origin of each QTL to the progenitors of the population; the corresponding results are presented in very informative figures. Finally, the paper is well structured and written, however, additional clarifications, justifications and possibly small additional analyses are requested to make the paper a better fit to the *eLife* community.

Major Comments:

1. Please justify the p-value thresholds:

– Lines 306-7 say that a 10fold weaker threshold is used for the interaction test--why? Also, where do these thresholds come from?

– Is it sound to take this two-step approach of first finding loci and then 'fine mapping' variants nearby? It is definitely fine if the thresholds are valid genome-wide--but then why not test genome-wide?

2. Please eliminate nonnegativity constraints on VCs in software:

– Are you truncating estimates to be nonnegative? If so, this will cause bias, which I believe is visible in your simulations. This is essential to fix for testing, and also for unbiased VC estimates.

– Users will test random effects, so it is necessary to validate (or remove) this part of your software.

– I don't see an assessment of tests or ses for the VCs. A quick supp figure corresponding to the current simulations would suffice, I think.

3. Please assess standard errors/false positive rates/power for testing VCs:

– Are there evidence that GxEMM improves power for fixed effect interaction tests? I think just commenting on this is enough, but it does seem important if you want others to use your approach. Comparing power (or some measure of false positives) for EMMA vs GxEMM in real data and/or simulation would work.

4. Mouse phenotyping:

– the text needs to say upfront that the mice included in this study were subject to a battery of phenotyping beyond weighing, including metabolic cage phenotyping, blood drwas, and "challenge-based phenotyping procedures". Currently this information comes late in the text (L287), and the title of Figure 1 Supplement 1 reads "Raw phenotype measurements", which does not alert the reader to these additional phenotypes. Some details about this phenotyping are needed, in particular how many blood draws were taken and which specific procedures are included under "challenge-based phenotyping".

This extra phenotyping means that the body weight results presented in this study could be study-specific. In particular, the decrease in PVE observed with age in this study, which contrasts with increased heritability with age observed in previous studies (see Introduction), could be the result of increased noise due to extra phenotyping. The implications of this extra phenotyping need to be discussed.

Importantly, it does not mean that this study is any less relevant than mouse studies without challenging life events, humans also experience challenging life events

5. Clarifications on Figure 4:

– What do PVE_tot and PVE_e represent (more on this below)? In particular, L291 "total genetic variance for the 40% CR group" is non-sensical to me because I thought PVE_tot was across all mice – as suggested by the fact that there is a single dark grey line in Figure 4 showing PVE_tot. Also, I was expecting PVE_tot (dark grey line in Figure 4) to be the sum of non diet-dependent genetic effects + all 5 PVE_e (colored lines). Yet the dark grey line is below the coloured lines. Hence I don't have the right intuition for what PVE_tot and PVE_e are. Looking at the equations didn't help: shouldn't it be PVE_e in Equation (7) and PVE_tot in Equation (13)?

Clarifications are needed.

– Figure 4: I love this figure, but worry a bit about the role of GxGeneration:

– The legend gave me a strong (and wrong) sense that the GxE was all about diet. If possible, disentangling the two GxE contributions visually would be helpful.

– I don't understand why generation is included (eg you could instead include cage effects, or neither).

– Why not have an apples-to-apples comparison using GxGeneration as the baseline for questions about GxDiet?

– What is dark grey? The hom estimate from GxEMM? The hom+GxGeneration (but then shouldn't there be one estimate for each generation)?

– More generally, I don't follow how generation is being used in this paper. Is it just always in the background as a random effect, and if so, why? Why don't you test for locus-x-generation effects--if they're not expected to be interesting, why do they explain so much variance? If they are absorbing confounding, why not also adjust for cage (perhaps with the GxEMM 'IID' model to save degrees of freedom)? I suspect whatever you're doing is reasonable, but maybe just explain a bit more clearly.

Figure 4—figure supplement 2, (b), right – isn't it concerning that the permuted phenotypes are heritable? How can this be? I recognize you don't use this kinship in your real analyses (though believe you do use the genotypes from which the kinship is built), but some explanation here would increase confidence in the overall approach.

I'd like to see versions of Figure 4 that: overlay the interventions, as in supplement Figure 1; only show the E-specific genetic variances (ie substracting the sig2g_hom terms); and show the heritabilities (ie normalize by total variance at each time point). These would make nice supplement Figures if easy to generate.

6. GWAS-related comments:

– The GWAS section and the corresponding figures are overwhelming. I think it would benefit from shortening and using sub-headings. I think it would be good to show first how likely causal variants can be identified (including FAP groups, functional annotations etc. and have one figure for that), with the chr6 and chr 12 loci as examples, and THEN show these loci across age and diets to discuss pleiotropy and heterogeneity.

– except if we consider triallelic SNPs, the distinction between allelic heterogeneity ("a single locus harboring multiple functional alleles each with distinct phenotypic effects") and (what I call locus heterogeneity) "a single genomic region contains multiple functional body weight loci that are only revealed with sufficient fine-mapping resolution" (L346-349) is difficult to make because noise can blur genetic associations, making it impossible to know the exact position of the causal variant(s). Consistent with the difficulty I just mentioned, in the section L419-443 and corresponding Supplementary Figures, I find that the assignments of the loci to either allelic or locus heterogeneity not obvious/robust.

I think it would be best to focus on distinguishing between pleiotropy (same causal variant(s) affecting body weight at different ages or in different diets) and non-pleiotropy (be it allelic or locus heterogeneity), rather than trying to distinguish between allelic and locus heterogeneity.

Statistical tests exist to test the null hypothesis {same locus affects both traits} in multiparental populations (doi.org/10.1534/g3.119.400098). In addition, looking at the FAPs of the lead variants, as the authors do here, would be a great way to complement this statistical test. Statistical evidence of different loci + differing top FAPS would provide strong evidence for some form of heterogeneity.

If you are going to focus on pleiotropy yes/no, it would be good to show a locus with likely pleiotropic effects (with the caveat that this would be the null hypothesis so not really evidence, rather suggestive).

An idea now, not a request: It would be interesting to know whether the loci for which different causal variants are predicted to affect different traits (e.g. different ages) are also loci for which multiple causal variants are predicted to affect either trait (see doi:10.1038/ng.2644).

7. Gene enrichment:

– is there a significant enrichment in genes affecting neurological behavior in this study, to support the claim that the fine-mapped genes implicate neurological and metabolic processes? If not, probably best to move the corresponding section of the text to the discussion. Also discuss whether the associations between neurological genes and body weight could arise due to the stressful extra phenotyping of the mice (are those associations mostly for ages after the challenge-based phenotyping for example, or are they seen even for ages before that stressful phenotyping?)

[Editors’ note: further revisions were suggested prior to acceptance, as described below.]

Thank you for submitting a revised version of your article "Age and diet shape the genetic architecture of body weight in Diversity Outbred mice" for consideration by *eLife*. Your article has been reviewed by 3 peer reviewers, and the evaluation has been overseen by a Reviewing Editor and Matt Kaeberlein as the Senior Editor. The following individuals involved in review of your submission have agreed to reveal their identity: Andrew Dahl (Reviewer #2); Amelie Baud (Reviewer #3).

The manuscript has improved since last submission, however, few comments many for clarity need to be addressed before the manuscript can be accepted for publication.

Essential Revisions:

1) The methods used to calculate the genome-wide significance threshold for the GWAS are now explained with sufficient detail but I believe the thresholds used are much less stringent than commonly accepted (commonly accepted = some estimate of a genome-wide threshold of 0.05). Two lines of evidence for this:

– The authors "used the number of top singular values that explain at least 90% of variation across markers (500), as a proxy for the number of independent QTL signals,", from which they calculated the Bonferroni corrected threshold. They mention in their response to the reviewers that this approach is similar to what is used in human GWAS. However, based on the results presented in Table 1 of https://onlinelibrary.wiley.com/doi/epdf/10.1002/gepi.20430 for example, I believe a value much greater than 90% of variation needs is used when applying this approach to human GWAS data to get close to the commonly accepted threshold of 5.10-8. SimpleM (presented in Table 1 of the paper above) uses a cutoff of 99.5% and still yields a more lenient threshold than 5.10-8.

– Other GWAS in DO mice have used much more stringent thresholds than p-value of 10-4: for example, https://link.springer.com/article/10.1007/s00335-018-9745-8#Sec11 used a P value threshold of 7.3.10-7 to claim genome-wide significance at 0.05; https://reader.elsevier.com/reader/sd/pii/S0168952519300654?token=DEFE149F2C71C5E4EA274AB6AE3638597EEACA678E9451C00AEE8CCD806BDA0E20ACE4BAA8DEF5D6BE37F274132E9F61&originRegion=eu-west-1&originCreation=20210602135101 used an even more stringent threshold – see Figure 3C.

As a consequence of the likely very lenient significance thresholds used in GWAS, all the claims from GWAS and subsequent analyses are supported by limited statistical evidence. In particular, it is difficult to be confident that the apparent lack of pleiotropic signals observed across loci is a real feature of the architecture of body weight: the different FAPs observed etc. could be the result of noise. The genes identified in Table 1 are also supported by limited statistical evidence.

2) I find the GWAS-related text still poorly organised and overwhelming. For example, effects as a function of age are discussed in three different sections (L404, L454, and L483). Similarly pleiotropy is discussed in three different sections, and functional variants are discussed in different sections too. Perhaps group everything related to pleiotropy in one section, effects as a function of age in another, and candidate variants/genes in a third?

The non-linearity in diet is hard to understand and appreciate in the current manuscript. It is quite interesting (surprising), if true. Perhaps give it its own section?

3) Using a diagonal omega matrix is problematic not only for pairs of diets but also for pairs of generations I think, as interaction effect sizes could well be correlated in pairs of generations. The response from the authors to this issue is not entirely satisfactory to me, as they acknowledge that biases are possible but suggest biases will be small without really explaining why they will be small.

– Lines 291-5 – I don't understand why permutations don't fully break the correspondence between both sources of relatedness and phenotype? The kinship is just a matrix, and I don't see how substructures in this matrix could be invariant to random permutation. (You are mean centering such that row/column sums are 0, right? this is required for REML). In other words, what null are you testing if the permutations are also significant? Why doesn't your claim imply that a pure noise phenotype would have significant GxE?

4) For negative VC estimates -- the new supp Figure is great, as is the new flag, but don't see these mentioned in the text? I suggest you acknowledge your constrained approach causes bias (cf your simulations), but that you address it in principle with the new flag and also there is no reason to worry about your real data analyses (because you have such strong signals). I think it's important to have unbiased inference for GxE h2 (much more than for additive h2).

---

## [Author Response]

Major Comments:1. Please justify the p-value thresholds:– Lines 306-7 say that a 10fold weaker threshold is used for the interaction test--why? Also, where do these thresholds come from?

The Bonferroni corrected p-value threshold used in the manuscript is a genome-wide threshold and was computed based on the expected number of linkage disequilibrium (LD) blocks (independent signals) in our study (similar to the strategy used in human GWAS studies). If the LD structure is perfectly block-like, the number of LD blocks can be computed from the number of non-zero singular values of the marker genotype matrix. In practice, we used the number of top singular values that explain at least 90% of variation across markers (500), as a proxy for the number of independent QTL signals, resulting in a p-value threshold of 10^-4^. This same threshold is reasonable for our interaction tests since we have only one test per marker (similar to the additive tests); however, we use a weaker threshold of 10^-3^ for interaction tests to explore suggestive GxE interaction QTLs. Our choice of thresholds, although arbitrary, are intended to help us focus our attention to the strongest QTLs with sufficient statistical support and are not intended to quantify the exact number of QTLs linked to body weight. We have updated the manuscript to better explain our choice of p-value thresholds.

– Is it sound to take this two-step approach of first finding loci and then 'fine mapping' variants nearby? It is definitely fine if the thresholds are valid genome-wide--but then why not test genome-wide?

The p-value thresholds we used are valid genome-wide and it is certainly reasonable to simply test genome-wide. We chose to take the two-step approach for two reasons:

1. At the genotyped markers, testing for association between founder-of-origin and phenotype has higher power than testing for association between allele count and phenotype (PMC4169154, PMC3149489). Notably, we only have informative founder-of-origin inferences at genotyped markers.

2. The two-step approach is much more computationally efficient. While this two-step approach certainly has an increased false-negative rate, evaluating the tests longitudinally through age should mitigate this substantially.

2. Please eliminate nonnegativity constraints on VCs in software:– Are you truncating estimates to be nonnegative? If so, this will cause bias, which I believe is visible in your simulations. This is essential to fix for testing, and also for unbiased VC estimates.– Users will test random effects, so it is necessary to validate (or remove) this part of your software.– I don't see an assessment of tests or ses for the VCs. A quick supp figure corresponding to the current simulations would suffice, I think.

Thank you for bringing to our attention the possibility that variance components do not need to be constrained to be positive, in order to have valid estimates of heritability. We re-computed the PVE estimates for all ages and diets without truncating the estimates to be non-negative, and found very little difference in the magnitude and trends of PVE with age (see Figure 4 —figure supplement 4). We chose to keep the estimates with non-negativity constraints as the main result, since these variance components are more interpretable in the context of our study. We have, however, included an option in the software to relax the non-negativity constraints on the variance components.

3. Please assess standard errors/false positive rates/power for testing VCs:– Are there evidence that GxEMM improves power for fixed effect interaction tests? I think just commenting on this is enough, but it does seem important if you want others to use your approach. Comparing power (or some measure of false positives) for EMMA vs GxEMM in real data and/or simulation would work.

To address the question of power for both fixed effect additive and interaction tests, we performed QTL mapping using the EMMA model, at the same set of ages as for the GxEMM model. In order to compare the GxEMM results to the standard linear mixed model, we reran the body weight association analyses at all ages using the EMMA model. We found that for both the additive and interaction models, at p-value thresholds of 10^-3^ and 10^-4^, the EMMA model identified more loci associated with body weight, quantified both by the number of independent genomic regions and average length of time in days per locus. We have updated the Results section, included Figure 5 —figure supplement 2 , and added a table in Supplementary file 3 to highlight this point.

We, however, do not claim the difference in QTLs to be due to a difference in power nor label any model-specific QTLs as false positives. Comparing Figure 5 —figure supplements 1 and 2, we observe that it is rare for a strong association under one model to be completely ablated under the other model; instead we observe subtle differences in likelihood ratio and p-values. Given the high temporal resolution in phenotype measurements, we feel it is more informative to focus on the effect size estimates (and longitudinal trends therein) of reasonably significant QTLs rather than focusing on the (admittedly arbitrarily and binary) problem of testing for QTLs. We relied on GxEMM since it is a more expressive model and EMMA is a special case of GxEMM.

4. Mouse phenotyping:– The text needs to say upfront that the mice included in this study were subject to a battery of phenotyping beyond weighing, including metabolic cage phenotyping, blood drwas, and "challenge-based phenotyping procedures". Currently this information comes late in the text (L287), and the title of Figure 1 Supplement 1 reads "Raw phenotype measurements", which does not alert the reader to these additional phenotypes. Some details about this phenotyping are needed, in particular how many blood draws were taken and which specific procedures are included under "challenge-based phenotyping".

This is a good point and we have updated the ‘Data’ section to describe the challenge-based phenotyping procedures (L129 – L134).

This extra phenotyping means that the body weight results presented in this study could be study-specific. In particular, the decrease in PVE observed with age in this study, which contrasts with increased heritability with age observed in previous studies (see Introduction), could be the result of increased noise due to extra phenotyping. The implications of this extra phenotyping need to be discussed.Importantly, it does not mean that this study is any less relevant than mouse studies without challenging life events, humans also experience challenging life events

Agreed; for this reason – and for all the unmeasured factors that we are unaware of – some of these results may be specific to this study. With data from another dietary intervention study in aged, outbred mice, we could assess whether this is the case. We have added text to the ‘Discussion’ to highlight this point (L557 – L559).

5. Clarifications on Figure 4:– What do PVE_tot and PVE_e represent (more on this below)? In particular, L291 "total genetic variance for the 40% CR group" is non-sensical to me because I thought PVE_tot was across all mice – as suggested by the fact that there is a single dark grey line in Figure 4 showing PVE_tot. Also, I was expecting PVE_tot (dark grey line in Figure 4) to be the sum of non diet-dependent genetic effects + all 5 PVE_e (colored lines). Yet the dark grey line is below the coloured lines. Hence I don't have the right intuition for what PVE_tot and PVE_e are. Looking at the equations didn't help: shouldn't it be PVE_e in Equation (7) and PVE_tot in Equation (13)?Clarifications are needed.

PVE_tot_ is computed using all mice (dark grey line in Figure 4) and PVE_e_ is computed using mice belonging to diet ‘e’ (colored lines in Figure 4). Since PVE_tot_ and PVE_e_ (for each diet) are computed on different sets of mice, we can’t expect PVE_tot_ to be the sum of all PVE_e_ (i.e., the denominators are different). This is clear from the equations: the denominator of Equation 7 for PVE_tot_ is Var_Y_ , the phenotypic variance computed using all the samples, while the denominator of Equation 13 for PVE_e_ is Var_Y|e_ , the variance computed using only samples belonging to diet ‘e’. Crucially, PVE_e_ is not the fraction of PVE_tot_ contributed by diet ‘e’.

Thank you for pointing out the error on L291; we have removed the term “total” from that sentence.

– Figure 4: I love this figure, but worry a bit about the role of GxGeneration:– The legend gave me a strong (and wrong) sense that the GxE was all about diet. If possible, disentangling the two GxE contributions visually would be helpful.– I don't understand why generation is included (eg you could instead include cage effects, or neither).– Why not have an apples-to-apples comparison using GxGeneration as the baseline for questions about GxDiet?– More generally, I don't follow how generation is being used in this paper. Is it just always in the background as a random effect, and if so, why? Why don't you test for locus-x-generation effects--if they're not expected to be interesting, why do they explain so much variance? If they are absorbing confounding, why not also adjust for cage (perhaps with the GxEMM 'IID' model to save degrees of freedom)? I suspect whatever you're doing is reasonable, but maybe just explain a bit more clearly.

The GxEMM model includes variance components for interaction terms with ‘diet’ and ‘generation’. In our study design, ‘generation’ is a proxy for the shared environment that the mice are exposed to throughout the study; in particular, mice of the same generation are subject to various phenotyping pipelines together. In this sense, ‘generation’ effects are essentially batch effects. We chose to always control for GxGeneration random effects in the model since this helped give us better and more precise estimates for PVE_e_. We don’t test for GxGeneration fixed effects, both because they are not expected to be interesting and because their contribution to diet-dependent PVE values are relatively small (see Figure 4 —figure supplement 5).

PVE_e_ is computed using mice on diet ‘e’ and includes contributions from the diet ‘e’ term and from all the ‘generation’ terms. Thus, PVE_e_ for two diets does have some shared components coming from the generation variance components; however, it is difficult to decompose which parts are truly shared and which are diet-specific, since the PVE are computed on different sets of mice, and thus the contribution of kinship is slightly different (see Equation 14). We have added Figure 4 —figure supplement 4 highlighting the contribution of diet-specific variance components and generation-specific variance components for each PVE_e_.

We could certainly control for GxCage effects, either assuming the same variance component for all cages as in the GxEMM ‘IID’ model or using a hierarchical model for the cage variance components. Assuming that all cages shared the same variance component parameter, we observed that the estimate for the cage-specific variance components was negligibly small, suggesting that either cage effects were negligible or the ‘IID’ model is too restrictive. A hierarchical model for cage effects is an interesting avenue for future work, but is out of the scope for our current study.

– What is dark grey? The hom estimate from GxEMM? The hom+GxGeneration (but then shouldn't there be one estimate for each generation)?

Dark grey in Figure 4 is the PVE computed using all mice. This includes the contribution from all the diet-specific and generation-specific variance components. We have included Figure 4 —figure supplement 5 showing the contribution of each of these components to the dark grey line.

Figure 4—figure supplement 2, (b), right – isn't it concerning that the permuted phenotypes are heritable? How can this be? I recognize you don't use this kinship in your real analyses (though believe you do use the genotypes from which the kinship is built), but some explanation here would increase confidence in the overall approach.

The SNP-based kinship can be decomposed into two components: the genetic sharing between the founder strains (the 8 founders have some degree of genetic similarity amongst themselves at the SNP level) and the genetic sharing arising from the breeding strategy used to develop the DO panel. In other words, SNP-based kinship accounts for sharing due to identity-by-state while FAP-based kinship accounts for sharing due to identity-by-descent. Randomly permuting the phenotype labels breaks the link between phenotypic similarity and the latter component of SNP-based kinship, but does not effectively break the link with the former component. Computing kinship using the founder allele probability effectively controls for any genetic similarity between the founder strains (i.e., it explicitly only includes the latter component). We believe this explains the inflation of PVE after permuting phenotypes with the SNP-based kinship, observed both for the EMMA and GxEMM models. Based on this reasoning, we hypothesize that the observed inflation would have been worse at earlier generations of DO mice and predict that this inflation will continue to decrease with future generations of DO mice. We have added text in the manuscript elaborating on this observation.

I'd like to see versions of Figure 4 that: overlay the interventions, as in supplement Figure 1; only show the E-specific genetic variances (ie substracting the sig2g_hom terms); and show the heritabilities (ie normalize by total variance at each time point). These would make nice supplement Figuresif easy to generate.

We have added Figure 4 —figure supplement 2, overlaying the interventions onto the PVE trends. We have also added Figure 4 —figure supplement 5 breaking down each PVE (total and diet-dependent) into contributions from the diet-specific terms and the generation-specific terms.

6. GWAS-related comments:– The GWAS section and the corresponding figures are overwhelming. I think it would benefit from shortening and using sub-headings. I think it would be good to show first how likely causal variants can be identified (including FAP groups, functional annotations etc. and have one figure for that), with the chr6 and chr 12 loci as examples, and THEN show these loci across age and diets to discuss pleiotropy and heterogeneity.

We appreciate the comments from this reviewer and agree that the GWAS section as originally constructed was overwhelming. To address this, we have now broken this section up into two subsections and eliminated the confusing discussion about distinguishing between pleiotropy, allelic heterogeneity and tightly linked loci (see below). We believe these changes will greatly increase reader comprehension.

In regards to figure 5, we agree there is a lot of information in each subpanel. We have re-organized these results into two figures to first discuss the identification of focal variants and second the effect of a focal variant on body weight at different ages.

– except if we consider triallelic SNPs, the distinction between allelic heterogeneity ("a single locus harboring multiple functional alleles each with distinct phenotypic effects") and (what I call locus heterogeneity) "a single genomic region contains multiple functional body weight loci that are only revealed with sufficient fine-mapping resolution" (L346-349) is difficult to make because noise can blur genetic associations, making it impossible to know the exact position of the causal variant(s). Consistent with the difficulty I just mentioned, in the section L419-443 and corresponding Supplementary Figures, I find that the assignments of the loci to either allelic or locus heterogeneity not obvious/robust.I think it would be best to focus on distinguishing between pleiotropy (same causal variant(s) affecting body weight at different ages or in different diets) and non-pleiotropy (be it allelic or locus heterogeneity), rather than trying to distinguish between allelic and locus heterogeneity.

Yes, this is a good point. We agree that in general we do not have a large enough mapping population for the fine-mapping analysis to distinguish between allelic and locus heterogeneity. We have updated the language in the manuscript to state that these results are not consistent with a single pleiotropic locus.

Statistical tests exist to test the null hypothesis {same locus affects both traits} in multiparental populations (doi.org/10.1534/g3.119.400098). In addition, looking at the FAPs of the lead variants, as the authors do here, would be a great way to complement this statistical test. Statistical evidence of different loci + differing top FAPS would provide strong evidence for some form of heterogeneity.If you are going to focus on pleiotropy yes/no, it would be good to show a locus with likely pleiotropic effects (with the caveat that this would be the null hypothesis so not really evidence, rather suggestive).

This is a great point. We do not apply this formal test of pleiotropy. We have updated the manuscript to state that when we identify a single FAP associated with body weight at multiple ages (e.g. Figures 6E,F; 7E) that this is suggestive of pleiotropy.

Additionally, we interpret the identification of distinct FAPs at the same genomic region (e.g. chromosome 6 locus with AJ/NOD and B6/CAST/PWK FAPs highlighted in Figures 5 and 6) as strong evidence against a single pleiotropic locus.

An idea now, not a request: It would be interesting to know whether the loci for which different causal variants are predicted to affect different traits (e.g. different ages) are also loci for which multiple causal variants are predicted to affect either trait (see doi:10.1038/ng.2644).

Yes, we agree this is an interesting idea to pursue in follow-up analyses.

7. Gene enrichment:– is there a significant enrichment in genes affecting neurological behavior in this study, to support the claim that the fine-mapped genes implicate neurological and metabolic processes? If not, probably best to move the corresponding section of the text to the discussion. Also discuss whether the associations between neurological genes and body weight could arise due to the stressful extra phenotyping of the mice (are those associations mostly for ages after the challenge-based phenotyping for example, or are they seen even for ages before that stressful phenotyping?)

We have not tested for a specific GO term enrichment of neurological and metabolic processes for the candidate genes. In response to this comment, we have edited this section of the results to make clear that we do not claim any specific enrichment of these processes. In keeping with comments from another reviewer, we feel that highlighting these six specific candidate genes (Negr1, Gphn, Trm1-like, Creb5, PI3K-C2.γ, and Edem3) adds significant value to the paper. Moreover, we believe that highlighting these six genes in the Results section immediately following our discussion of the fine-mapping data is the clearest way to convey this information to the reader and we have kept this in the Results section.

We agree with this comment (and previous comments) that this particular result, and nearly any other result we present in the paper could in fact be caused by peculiarities of this experiment rather than the ‘true’ genetic basis of variation in body weight and how that is affected by age and diet. We have emphasized this caveat in the Discussion. Future studies that incorporate longitudinal measurements of body weight in both mice and humans should help identify how much of our results are robust to differences in experimental design.

[Editors’ note: further revisions were suggested prior to acceptance, as described below.]

Essential Revisions:1) The methods used to calculate the genome-wide significance threshold for the GWAS are now explained with sufficient detail but I believe the thresholds used are much less stringent than commonly accepted (commonly accepted = some estimate of a genome-wide threshold of 0.05). Two lines of evidence for this:– The authors "used the number of top singular values that explain at least 90% of variation across markers (500), as a proxy for the number of independent QTL signals,", from which they calculated the Bonferroni corrected threshold. They mention in their response to the reviewers that this approach is similar to what is used in human GWAS. However, based on the results presented in Table 1 of https://onlinelibrary.wiley.com/doi/epdf/10.1002/gepi.20430 for example, I believe a value much greater than 90% of variation needs is used when applying this approach to human GWAS data to get close to the commonly accepted threshold of 5.10-8. SimpleM (presented in Table 1 of the paper above) uses a cutoff of 99.5% and still yields a more lenient threshold than 5.10-8.– Other GWAS in DO mice have used much more stringent thresholds than p-value of 10-4: for example, https://link.springer.com/article/10.1007/s00335-018-9745-8#Sec11 used a P value threshold of 7.3.10-7 to claim genome-wide significance at 0.05; https://reader.elsevier.com/reader/sd/pii/S0168952519300654?token=DEFE149F2C71C5E4EA274AB6AE3638597EEACA678E9451C00AEE8CCD806BDA0E20ACE4BAA8DEF5D6BE37F274132E9F61&originRegion=eu-west-1&originCreation=20210602135101 used an even more stringent threshold – see Figure 3C.As a consequence of the likely very lenient significance thresholds used in GWAS, all the claims from GWAS and subsequent analyses are supported by limited statistical evidence. In particular, it is difficult to be confident that the apparent lack of pleiotropic signals observed across loci is a real feature of the architecture of body weight: the different FAPs observed etc. could be the result of noise. The genes identified in Table 1 are also supported by limited statistical evidence.

We appreciate the reviewers pointing out evidence from both human GWAS and mouse QTL studies suggesting the need for a more stringent p-value cutoff for claiming significance. While the two papers on GWAS in DO mice used a more stringent p-value threshold to claim genome-wide significance at 0.05, the Yuan et al., paper used a different permutation scheme to generate their null distribution and compute their p-value threshold and the Saul et al., paper does not describe how their threshold was selected.

Following the approach in the Gao et al., paper on Bonferroni correction in human GWAS studies, we computed the eigenvalues of the full linkage disequilibrium matrix among genotyped variants and used the top K eigenvalues that explain Y% of total genetic variation as a proxy for the number of independent tests. For Y = 90, we obtained K = 500, leading to a p-value threshold of 1e-4. For Y = 99.5, we obtained K = 4455, leading to a p-value threshold of 1.1e-5. A Bonferroni correction for multiple hypotheses is considered to be highly conservative, particularly for traits like Body Mass that are known to have a strong genetic component. An alternate and less conservative approach would be to use a p-value threshold corresponding to a fixed, expected number of false discoveries. Using the Benjamini-Hochberg method, a procedure that is expected to be robust to correlations due to linkage disequilibrium, we computed a p-value threshold of 1.2e-4 corresponding to a 10% expected false discovery rate.

Given the high heritability of Body Mass and the availability of longitudinal Body Mass data, we reasoned that it was excessively conservative to rely on a single Bonferroni-based p-value threshold as the only source of statistical evidence. Instead, in Table 1, we listed loci that (i) passed a more liberal Bonferroni threshold (matching the 10% FDR threshold), (ii) were identified as significant at multiple ages, and (iii) were found to be significant using fine-mapping analysis. In order to clarify the strength of evidence based on p-values alone, we marked in Table 1 loci with p-value less than 1.1e-5 with *** (denoting them as highly significant), loci with p-value less than 1e-4 with ** (denoting them as significant), and p-value less than 1e-3 with * (denoting them as suggestive). Multiple sources of evidence supporting these loci suggest that the observed associations and related aspects of the architecture of body weight are not the result of noise.

2) I find the GWAS-related text still poorly organised and overwhelming. For example, effects as a function of age are discussed in three different sections (L404, L454, and L483). Similarly pleiotropy is discussed in three different sections, and functional variants are discussed in different sections too. Perhaps group everything related to pleiotropy in one section, effects as a function of age in another, and candidate variants/genes in a third?The non-linearity in diet is hard to understand and appreciate in the current manuscript. It is quite interesting (surprising), if true. Perhaps give it its own section?

We have restructured the GWAS-related text to improve readability and organization of information. First, we discussed the use of founder allele patterns, along with functional variants and candidate genes at the two loci we’ve highlighted, then we discussed pleiotropy, and, finally, we discussed effects as a function of age. We have retained a separate, broader discussion on candidate genes that we’ve identified in this study. We have also separated the paragraph on non-linearity in effects with diet into its own section.

3) Using a diagonal omega matrix is problematic not only for pairs of diets but also for pairs of generations I think, as interaction effect sizes could well be correlated in pairs of generations. The response from the authors to this issue is not entirely satisfactory to me, as they acknowledge that biases are possible but suggest biases will be small without really explaining why they will be small.

We acknowledge the lack of explanation behind our intuition that off-diagonal terms in the omega matrix are expected to be small, leading to small biases in estimates of heritability. To address this, we modified our code to estimate a full omega matrix and computed heritability estimates at each age using the full matrix. We observed that off-diagonal entries of the full omega matrix were often orders of magnitude smaller than the diagonal entries, leading to very small changes in heritability estimates using the full matrix. We have added Figure 4 —figure supplement 5 comparing the heritability estimates between a diagonal omega matrix and a full omega matrix.

– Lines 291-5 – I don't understand why permutations don't fully break the correspondence between both sources of relatedness and phenotype? The kinship is just a matrix, and I don't see how substructures in this matrix could be invariant to random permutation. (You are mean centering such that row/column sums are 0, right? this is required for REML). In other words, what null are you testing if the permutations are also significant? Why doesn't your claim imply that a pure noise phenotype would have significant GxE?

The kinship is a matrix of relatedness between pairs of individuals; in this sense, substructures in this matrix (or even the entire kinship matrix) could be invariant to certain permutations of samples depending on whether kinship is computed using genotypes or using founder allele probabilities. This invariance is much more readily observed in the DO mouse population (or other lab-outbred populations) than in natural populations of mice or humans, particularly because the DO population is currently only ~30 generations removed from the inbred Collaborative Cross mouse lines from which they were derived.

As an extreme example, if the mice in our study were composed of just “copies” of the 8 founder strains, then the genotype-based kinship matrix would have eight blocks of matrix-of-ones along the diagonal (since the 8 founders are completely homozygous), with off-diagonal blocks capturing the kinship between pairs of founders. Such a kinship matrix is invariant under certain permutations of sample ids; these permutations would give significant heritability estimates and, thus, straightforward permutations would not be a valid approach to construct a null of no association between genotype and phenotype.

The current DO population matches a less extreme version of the example above, due to ~30 generations of outbreeding. Thus, permuting the genotype-based kinship matrix does not fully break the association between genotype and phenotype. We believe this effect would have been more extreme in earlier DO generations, and will continue to weaken in later DO generations.

4) For negative VC estimates -- the new supp Figure is great, as is the new flag, but don't see these mentioned in the text? I suggest you acknowledge your constrained approach causes bias (cf your simulations), but that you address it in principle with the new flag and also there is no reason to worry about your real data analyses (because you have such strong signals). I think it's important to have unbiased inference for GxE h2 (much more than for additive h2).

We have now addressed in the manuscript the possible bias in our constrained approach, shown that we observed little difference with and without the constraint, and highlighted that our software allows for relaxing this constraint via a flag.